# Failure of the Downstream Shoulder of Rockfill Dams Due to Overtopping or Throughflow



Ricardo Monteiro-Alves [1],*, Miguel Á. Toledo [1], Rafael Moran [1,2],* and Luis Balairón [3]

1   Civil Engineering Department: Hydraulics, Energy, and Environment, E.T.S. de Ingenieros de Caminos, Canales y Puertos, Universidad Politécnica de Madrid, 28040 Madrid, Spain; miguelangel.toledo@upm.es

2   International Centre for Numerical Methods in Engineering, Universitat Politècnica de Catalunya, Campus Norte, 08034 Barcelona, Spain

3   Hydraulics Laboratory, Centro de Estudios Hidrográficos, Centro de Estudios y Experimentación de Obras Públicas (CEDEX), 28005 Madrid, Spain; luis.balairon@cedex.es

*   Correspondence: ricardo.monteiro@upm.es (R.M.-A.); r.moran@upm.es (R.M.)

**Abstract:** This paper presents the results of an extensive laboratory set of tests aimed to study the failure of the downstream shoulder of highly permeable rockfill subjected to overflow. The experimental research comprised testing 114 physical models by varying the following elements: (i) the median size of the uniform gravels (7 to 45 mm); (ii) the configuration of the dam, i.e., upstream and downstream shoulders and crest or just the downstream shoulder; (iii) the dam height (from 0.2 to 1 m), (iv) the crest length (from 0.4 to 2.5 m), (v) the downstream slope (from 1 to 3.5 H:V), (vi) the type of impervious element (i.e., central core, upstream face, and no impervious element). The tests allowed us to identify two failure mechanisms, slumping and particle dragging. In addition, the downstream slope was observed to be one of the most important variables in this parametric study, as it influenced the pore water pressures inside the dam, the failure discharge, and the occurrence of one or the other mechanism of failure.

**Keywords:** rockfill dam; overtopping; dam failure; overflow; dam safety; floods; dam breach





## 1. Introduction

Rockfills may be formed by natural processes or as a direct result of human action in civil engineering structures. Examples of natural rockfills are moraine dams [1] and landslide or avalanche dams [2–4]. On the other hand, constructed rockfill structures include levees, dikes, and dams built to fulfill different human needs, and embankment-like deposits of homogeneous coarse rockfill, usually produced by mining activities, also referred to as rock drains [5].

The two main causes of failure of large rockfill dams registered up to 1986, excluding dams constructed in Japan pre-1930 and in China, are overtopping (55.6% of cases corresponding to 5 failures) and piping (11.1% of cases corresponding to 1 failure) [6]. Due to the high permeability of clean rockfill, both overtopping and piping lead to the formation of a seepage profile at the base of the downstream shoulder [7–18], that finally exits the dam at the toe [5,9,19–22]. In the toe, delimited upstream by the first emergence point [10,20], the hydraulic gradients and seepage forces are maximum and, besides that, point outward of the dam [22], making this area prone to erosion and a zone of primary engineering concern [5,19]. As a consequence, failure initiates at the toe for a discharge that must overcome a given threshold [3,23–26] and may occur by slumping, internal migration of particles, or surface unraveling erosion resulting in concentrated flow paths [22,26–30].

In natural rockfill dams, formed without any impervious element, the prediction of the final breach geometry and dimensions is crucial for the estimation of the peak outflow. Diverse studies have been performed in this area for both cohesive and non-cohesive materials [31–38], as well as literature reviews [39–41]. In dams constructed with an

impervious element, prediction of the breach geometry and dimensions is also important but, in this case, the stability of this element (internal core or upstream face), which becomes unprotected after the removal of the downstream shoulder, must also be analyzed. In these cases, the failure of the impervious element controls the breach hydrograph [42–47]. Either way, knowing the discharge that completes the failure of the downstream rockfill shoulder, i.e., the 'failure discharge' ($Q_f$) whereby damages to the downstream shoulder reach the crest, is relevant to understanding how far the dam is from a catastrophic failure.

Failure progression, patterns, and mechanisms are affected by the dam's geometric characteristics and material gradings [3]. Based on an extensive laboratory test campaign, this paper provides a parametric analysis to understand how the failure discharge is affected by some characteristics of the dam body. Based on the results of the parametric analysis, an empirical formulation is calibrated to estimate the 'failure discharge'.

## 2. Methodology

### 2.1. Test Overview

A total of 114 physical models (PM), all tested in horizontal flumes with rectangular sections, are summarized in Table 1. They were tested by varying the following elements: (i) the size of the uniform gravel, characterized by its D50; (ii) the configuration of the cross-section, using partial (PPM) or complete physical models (CPM) (complete configurations are trapezoidal and include both upstream (USS) and downstream shoulders (DSS) and crest, while partial configurations are triangular and include only the downstream shoulder); (iii) geometrical parameters as the height ($H$) of the physical models, the width of the flume ($W$), the width of the crest ($l_c$), the downstream and upstream slopes ($Z_{dss}$ and $Z_{uss}$, respectively); (iv) the type of impervious element (IE) (central core (CC), upstream face (UF), and no impervious element (NIE)).

**Table 1.** Summary of the 114 physical models tested to study the failure of the rockfill downstream shoulder (values in parentheses represent the number of models tested for that particular configuration). NA means 'not available'.

| $H$ (m) | $W$ (m) | $Z_{dss}$ (H:V) | $Z_{uss}$ (H:V) | $l_c$ (m) | IE | Material |
|---|---|---|---|---|---|---|
| 0.229 (4 models) | 0.4 (4/4) | 1.5 (1/4) 2.5 (1/4) 3.5(2/4) | 1.5 (4/4) | 0.057 (4/4) | UF (4/4) | M3 (4/4) |
| 0.5 (44 models) | 0.4 (2/44) 0.6 (25/44) 1.32 (11/44) 2.46 (6/44) | 1.0 (2/44); 1.1 (1/44) 1.3 (1/44); 1.4 (1/44) 1.5 (5/44); 1.6 (2/44) 1.75 (3/44); 1.9 (1/44) 1.95 (1/44); 2.0 (4/44) 2.1 (2/44); 2.2 (3/44) 2.25 (1/44); 2.3 (1/44) 2.4 (1/44); 2.5 (1/44) 2.6 (2/44); 2.7 (2/44) 2.75 (1/44); 2.8 (2/44) 2.9 (2/44); 3.0 (3/44) 3.1 (1/44); 3.3 (1/44) | NA (42/44) 1.5 (2/44) | NA (42/44) 0.1 (2/44) | NIE (38/44) CC (6/44) | M4 (22/44) M6 (2/44) M7 (20/44) |
| 0.6 (4 models) | 2.5 (4/4) | 1.5 (1/4) 1.75 (1/4) 2.5 (1/4) 2.7 (1/4) | 1.5 (4/4) | 0.2 (4/4) | NIE (4/4) | M2 (4/4) |
| 0.8 (3 models) | 1.0 (3/3) | 1.5 (1/3) 2.5 (1/3) 3.5 (1/3) | 1.5 (3/3) | 0.2 (3/3) | UF (3/3) | M7 (3/3) |

**Table 1.** *Cont.*

| $H$ (m) | $W$ (m) | $Z_{dss}$ (H:V) | $Z_{uss}$ (H:V) | $l_c$ (m) | IE | Material |
|---|---|---|---|---|---|---|
| 1 | 1.0 (37/61) | 1.5 (18/61) | NA (2/61) | NA (2/61) | NIE (34/61) | M1 (18/61) |
| (61 models) | 1.32 (2/61) | 1.6 (1/61) | 1.5 (59/61) | 0.2 (59/61) | CC (12/61) | M4 (13/61) |
| | 1.5 (2/61) | 1.9 (1/61) | | | UF (15/61) | M5 (3/61) |
| | 2.46 (17/61) | 2.2 (20/61) | | | | M6 (13/61) |
| | 2.5 (3/61) | 2.5 (2/61) | | | | M7 (12/61) |
| | | 3.0 (19/61) | | | | M8 (2/61) |

All of these variables combined can be grouped into four types of physical models as shown in Figure 1. These groups are (i) complete configuration with upstream face (CPM/UF), (ii) complete configuration without impervious element (CPM/NIE), (iii) partial or complete configuration with a central core (PPM/CC), and (iv) partial configuration without impervious element (PPM/NIE).

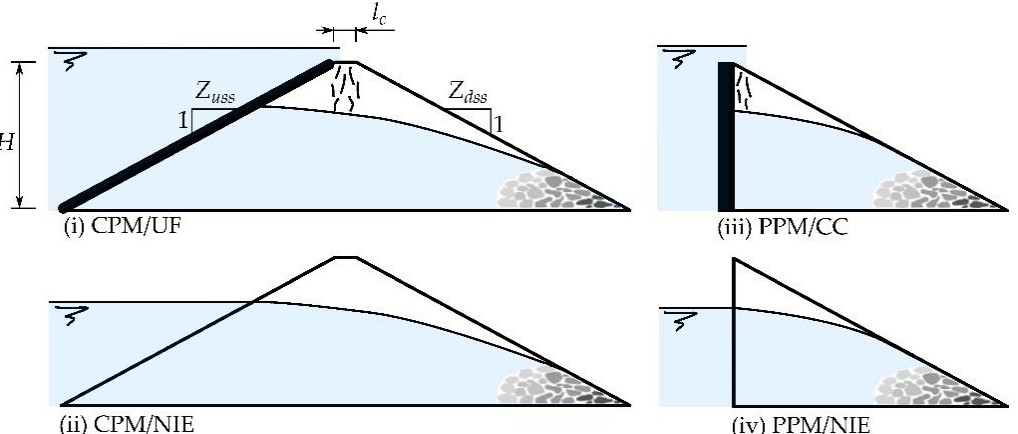

**Figure 1.** Schematic drawing of the different types of physical models tested at the laboratory: (**i**) Complete Physical Models with Upstream Face, (**ii**) Complete Physical Models with No Impervious Element, (**iii**) Partial Physical Models with Central Core and (**iv**) Partial Physical Models with No Impervious Element.

*2.2. Specific Tests*

Within the main campaign, we performed specific tests to evaluate the variability of the results and the scale effect. Regarding variability, five groups of tests were performed (from A to E), all CPM/NIE, consisting of repeating the same physical model a given number of times to assess if the test procedure could substantially affect the 'unit failure discharge' ($q_f$) or any other factors, such as the hydraulic pressures inside the downstream shoulder or the 'failure path', i.e., the evolution of the failure progress with the throughflow discharge [25,48]. It must be noted, though, that the discharge steps were not (in general) the same throughout the tests within the same group. Tests in the same group all had the same geometry and dimensions as well as the same granular material:

- [Group A]: $W = 2.50$ m, $H = 1$ m, $Z_{dss} = 2.2$, $Z_{uss} = 1.5$, Gravel M5 (tests 5, 6, 7).
- [Group B]: $W = 2.46$ m, $H = 1$ m, $Z_{dss} = 3.0$, $Z_{uss} = 1.5$, Gravel M7 (tests 8, 9).
- [Group C]: $W = 1.00$ m, $H = 1$ m, $Z_{dss} = 2.2$, $Z_{uss} = 1.5$, Gravel M6 (tests 81, 82, 85, 87, 88).
- [Group D]: $W = 1.00$ m, $H = 1$ m, $Z_{dss} = 1.5$, $Z_{uss} = 1.5$, Gravel M6 (tests 89, 90, 91, 92).
- [Group E]: $W = 1.00$ m, $H = 1$ m, $Z_{dss} = 3.0$, $Z_{uss} = 1.5$, Gravel M6 (tests 94, 95).

Regarding the scale effect, the aim was to analyze if the Froude similitude could be applied to scale $q_f$. So, for a scale factor $s_L = 1 : 3.5$, we tested 0.23 m high 'small-scale' physical models (tests nº 109, 111, and 130) and 0.8 m high 'prototypes' (tests nº 108, 110, and 112). This scale factor was applied to all lengths (except for the flume width) including the gravels' D50. So, gravels M3 and M7 were respectively used in the 'small-scale' model

and 'prototype'. In total, we tested three $Z_{dss}$ = 1.5, 2.5, and 3.5, and for each of these slopes, we tested one 'small-scale' physical model and a 'prototype'.

### 2.3. Test Setup and Procedure

The physical models were constructed by pouring and extending the granular material without compaction. Nevertheless, some unintentional compaction resulted from walking over the models during construction, mainly those constructed in the larger flumes involving the placement of tons of material. To obtain the final geometry, the physical model surfaces were evened with an aluminum straight guide.

Tests were based on a stepwise flow increment methodology until total failure of the downstream shoulder occurred. By total failure, we mean that the damages inflicted to the downstream shoulder reached the crest of the dam. Although the location of the upstream impervious face or the internal clay core was slightly different, this criterion permitted a homogeneous analysis of the results arising from tests with different types of impervious elements. The failure forefront (Figure 2) is the border that separates intact areas of the slope from those damaged by failure. The maximum advance of failure ($B_f$), i.e., the most upward point of the failure forefront, was used to define the complete failure of the shoulder. So, the physical models were defined as completely failed when $B_f$ reached the downstream edge of the crest. For practical purposes, the discharge that produced complete failure ($Q_f$) was defined as being the average value between the highest discharge in which failure did not reach the crest, and the lowest in which failure surpassed it.

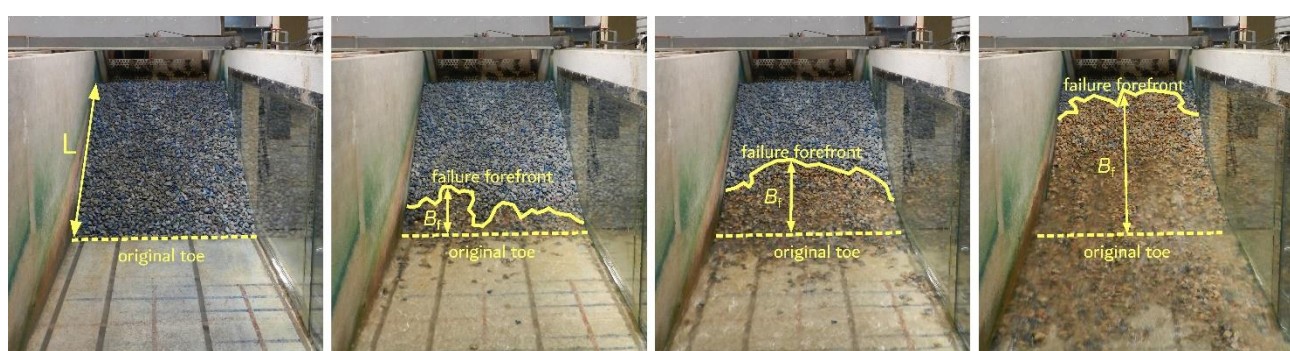

**Figure 2.** Scheme of the failure process sequence for discharges of 0, 0.0188, 0.0256, and 0.0423 $m^3\ s^{-1}$ (test nº 133, $H$ = 1 m, $W$ = 1.32 m, $Z_{dss}$ = 1.6, gravel M7).

Each discharge was kept constant until steady-state conditions were reached, i.e., until no additional damage was observed to the shoulder or any change in the water elevation and pressures. Several long preliminary tests (more than 1 h per step) showed that a step duration of 30 min was long enough for reaching the stationary state. The number of steps varied from test to test and, following the initiation of failure (first damage observed on the shoulder), five steps were performed, on average. The minimum and the maximum number of flow steps were three and ten, respectively. Once the stationary state condition of every discharge step was reached, all measurements were performed.

### 2.4. Materials

Tests were performed with eight uniform limestone gravels (M1 to M8) of different sizes, ranging D50 from 0.00736 to 0.04509 m, and with a coefficient of uniformity ($C_u$ = D60/D10) ranging from 1.46 to 2.28. Their main characteristics are summarized in Table 2 and Figure 3. These eight gravels were obtained by sieving four raw gravels with size ranges 4–12, 12–20, 20–40, and 40–80 mm. Materials M1 to M3 resulted from gravel 4–12 mm, M4 to M6 from gravel 12–20 mm, M7 from gravel 20–40 mm, and M8 from gravel 40–80 mm.

**Table 2.** Summary of the main geotechnical characteristics of the gravels used in the construction of the physical models. NA means 'not available'.

| Raw Materials | | | | |
|---|---|---|---|---|
| Variable | 4–12 mm | 12–20 mm | 20–40 mm | 40–80 mm |
| D10 particle size [mm] | 5.27 | 10.18 | 20.05 | NA |
| D50 particle size [mm] | 8.5 | 15.3 | 26.5 | NA |
| Coefficient of uniformity ($C_u$) | 1.75 | 1.60 | 1.40 | NA |
| Fine percentage (%) | 0.80 | 0.10 | 0.95 | NA |
| Specific gravity ($G$) | 2.70 | 2.70 | 2.70 | NA |
| Dry unit weight ($\gamma_d$) [kN·m$^{-3}$] | 14.7 | 14.5 | 15.0 | NA |
| Saturated unit weight ($\gamma_d$) [kN·m$^{-3}$] | 18.9 | 18.5 | 19.0 | NA |
| Porosity ($n$) [%] | 42.3 | 41.1 | 41.0 | NA |
| Void ratio ($e$) [%] | 73.5 | 69.5 | 66.0 | NA |
| Coefficient of permeability ($k$) [m·s$^{-1}$] [†] | 0.0008 | 0.0016 | 0.0051 | NA |
| Internal friction angle ($\varphi'$) [degrees] | 43.98 | 48.85 | 53.86 | NA |

| Sieved Materials | | | | | | | | |
|---|---|---|---|---|---|---|---|---|
| Variable | M1 | M2 | M3 | M4 | M5 | M6 | M7 | M8 |
| D10 particle size [mm] | 4.89 | 5.97 | 5.72 | 8.36 | 11.11 | 10.62 | 23.68 | 21.58 |
| D50 particle size [mm] | 7.36 | 8.20 | 9.98 | 12.64 | 16.49 | 17.33 | 35.04 | 45.09 |
| Coefficient of uniformity ($C_u$) | 1.63 | 1.46 | 1.87 | 1.54 | 1.58 | 1.80 | 1.56 | 2.28 |
| Specific gravity ($G$) | NA | NA | 2.60 | NA | NA | NA | 2.50 | 2.60 |
| Dry unit weight ($\gamma_d$) [kN·m$^{-3}$] | NA | NA | 14.7 | 16.1 | NA | 15.7 | 14.5 | 15.0 |
| Saturated unit weight ($\gamma_d$) [kN·m$^{-3}$] | NA | NA | 18.9 | 19.9 | NA | 19.7 | 18.5 | 19.0 |
| Porosity ($n$) [%] | NA | NA | 42.6 | 39.3 | NA | 40.8 | 41.2 | 41.0 |
| Void ratio ($e$) [%] | NA | NA | 73.5 | 64.7 | NA | 68.9 | 69.5 | 66.0 |
| Angle of repose ($\varphi_{repose}$) [degrees] | NA | NA | NA | 36.9 | NA | NA | 40.4 | 42.8 |
| Resistance law term $a$ [s·m$^{-1}$] | NA | NA | 1.44 | 2.71 | NA | 1.53 | 0.82 | 0.65 |
| Resistance law term $b$ [s$^2$·m$^{-2}$] | NA | NA | 144.77 | 65.35 | NA | 84.66 | 52.82 | 16.96 |

[†] These values were obtained for hydraulic gradients of 0.54 (4–12 mm), 0.20 (12–20 mm), and 0.06 (20–40 mm).

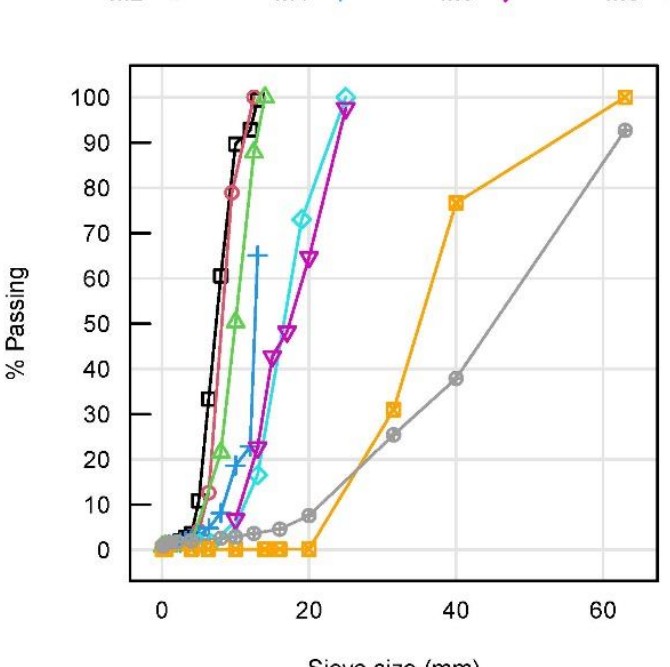

**Figure 3.** Particle size distribution of gravels from M1 to M8.

Some characterization tests were consigned to an external laboratory, the Geotechnics Laboratory of CEDEX, to obtain the particle size distribution (UNE EN 933-1), the specific gravity (*G*) of the soil solids (ASTM D5550-06), and the soil density (UNE 103301:1994) of gravels M3, M7, and M8. With these two last standards, we obtained the void ratio (*e*) with Equation (1), porosity (*n*) with Equation (2), and the saturated specific weight ($\gamma_{sat}$) with Equation (3). The particle size distribution was obtained for the rest of the materials following the same standards. The porosity of gravels M4 and M6 was obtained by filling a bucket full of gravel with water. The quadratic resistance law of flow through coarse granular materials expressed by Equation (4) relating the hydraulic gradient (*i*) to the flow velocity (*v*) was obtained for gravels M4, M7, and M8 following a methodology described in the state of the art [10,49], and for gravels M3 and M6 with a horizontal permeameter of large dimensions [50].

$$\gamma_{d} = \frac{G \cdot \gamma_{w}}{1 + e} \tag{1}$$

$$n = \frac{e}{1 + e} \tag{2}$$

$$\gamma_{sat} = \gamma_{d} + n \cdot \gamma_{w} \tag{3}$$

$$i = av + bv^2 \tag{4}$$

The angle of repose was obtained for gravels M4, M7, and M8 by scanning the surface of the mounds of these gravels using a 2D laser (LMS200-30106 by SICK$^{TM}$, scanning range for objects up to 10 m; angular range up to 180° with a maximum angular resolution of 0.25°; a systematic error of $\pm 0.015$ m; a statistical error of $\pm 0.005$ m). The repose angles were obtained by fitting a linear regression to the external surface of the mounds [45]. The Geotechnics Laboratory of CEDEX was also consigned to perform the characterization of the first three raw gravels to obtain the particle distribution (UNE 103101:1995), soil density (UNE 103301:1994), permeability (UNE 103403:1999), friction angles (UNE 103401:1998), and specific gravity of soil solids using a gas pycnometer (ASTM D5550-06).

### 2.5. Facilities and Instrumentation

Tests were conducted in four U-shaped flumes (rectangular section) located in two different laboratories: one flume at the Hydraulics Laboratory of the *E.T.S.I. de Caminos, Canales y Puertos* of the *Universidad Politécnica de Madrid* (UPM), and three at the Hydraulics Laboratory of the *Centro de Estudios Hidrográficos* of the *Centro de Estudios y Experimentación de Obras Públicas* (CEDEX), both laboratories located in Madrid (Spain).

The UPM flume, straight with horizontal bottom, is 13.7 m long, 2.5 m wide, and 1.3 m high (inner dimensions) with an inspection window 4.6 m long and 1.1 m high placed in the left-side wall (Figure 4a). In this flume, we tested physical models with different widths ranging from 0.6 to 2.5 m; hence, when a smaller width had to be tested, it was necessary to build a longitudinal central wall. Figure 4b,c show images of a test performed at UPM.

This flume was supplied using a constant water level tank (in which the water level was kept constant employing a pump with a variable-frequency drive) connected to the flume through a pipe 0.3 m in diameter with a manual/automated valve. This system could supply approximately 0.080 m$^3$s$^{-1}$ with this valve fully opened. An extra hydraulic pump, connecting directly the underground main tank with the flume through a different pipe, 0.5 m in diameter, could supply a constant inflow of up to approximately 0.120 m$^3$s$^{-1}$. Flows were measured downstream of the physical models using a sharp rectangular weir with lateral contraction (crest length and height were 0.502 and 0.28 m, respectively) located in the 0.8 m wide flume that returned water to the underground main tank (270 m$^3$ capacity through an area of 180 m$^2$). The water level upstream of the weir was measured with a P8000 ultrasonic sensor with a digital display (Dr. D. Wehrhahn, Hannover, Germany) measuring between 0.07 and 2 m with an accuracy of $\pm 0.0001$ m), located 0.69 m from the weir. The records of the water level were obtained visually by registering the displayed values. The hydraulic pressures were measured with a set of 84 piezometers spread over

seven transversal rows and twelve longitudinal lines (Figure 4a). Measurements were obtained by visual inspection using a millimetric ruler.

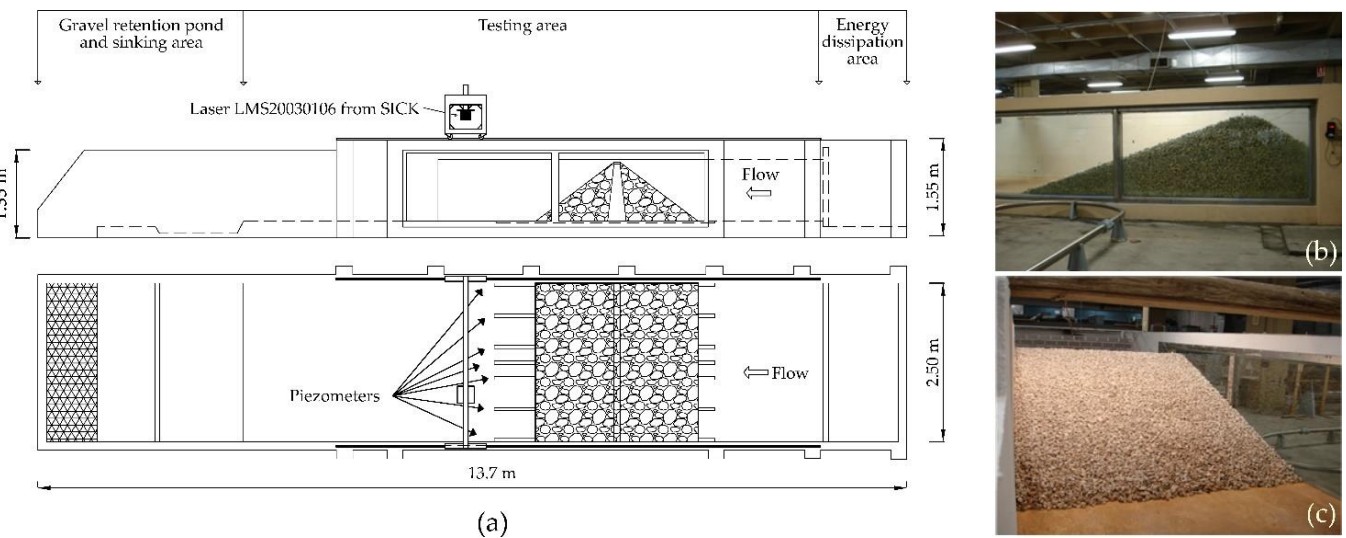

**Figure 4.** The UPM flume. (**a**) Scheme of the flume dimensions and areas, (**b**) image of a test taken from the outside of the flume through the inspection window located on the left-side wall, and (**c**) taken from inside the flume.

At the CEDEX laboratory, three flumes were used (Figure 5). The smaller one was a tilting metallic flume 12 m long, 0.4 m wide, and 0.6 m high. Although the slope of the flume could be controlled, it was kept horizontal throughout the tests. This flume was supplied by a constant water level tank where discharges were measured upstream of the physical model using a thin-plate rectangular weir 0.487 m long with no lateral contraction. This constant level tank was supplied with water by pumping from the main tank located below the laboratory floor (3000 m$^3$) with a Worthington hydraulic pump (three-phase motor GEAL 220/380 V, 4.4 kW, 6 hp) capable of pumping up to 0.06 m$^3$ s$^{-1}$ with 5 mwc. The medium-size concrete flume was 12 m long, 1.0 m wide, and 1.1 m high. Supplied by a constant water level tank, discharges were also measured upstream of the physical model with a 90° thin-plate triangular weir. The water level upstream of the weirs was measured with P8000 ultrasonic level sensors with digital displays (Dr. D. Wehrhahn, Hannover, Germany). This constant water level tank was also supplied with water from the main tank with a Worthington hydraulic pump (three-phase motor Alcanza 220/380 V, 45 kW, 6 hp) capable of pumping up to 0.2 m$^3$ s$^{-1}$ with 14 mwc. In this flume, the hydraulic pressures were measured using the intelligent pressure instrumental system Scanivalve, placed at the base of the flume. This system was composed of 44 measuring points distributed in ten transversal rows along a distance of 2.9 m. Finally, the bigger metal-glazed flume, 100 m long, 1.5 m wide, and 1.5 m high, was supplied directly from the main tank using two Jeumont-Schneider hydraulic pumps, one capable of pumping up to 1.7 m$^3$ s$^{-1}$ with 4.4 mwc (DC motor 107 kW and 440 V), and the other capable of pumping up to 0.8 m$^3$ s$^{-1}$ with 4.25 mwc (DC motor 55 kW and 440 V).

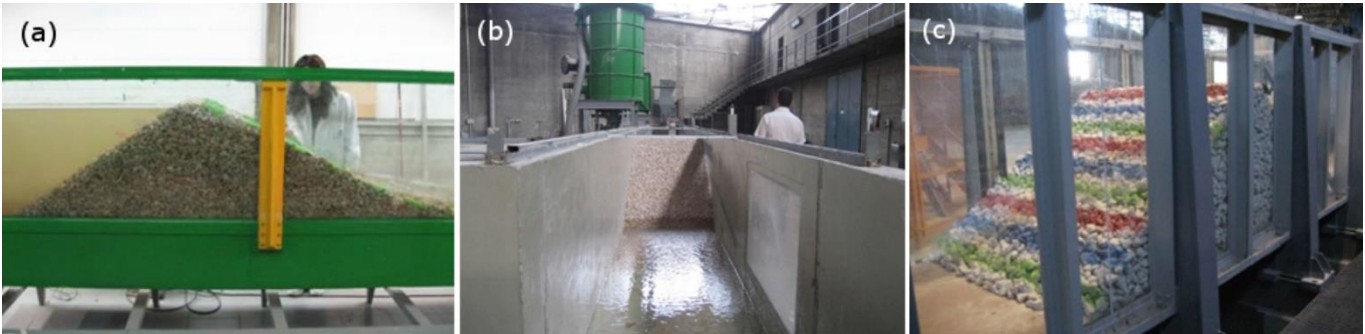

**Figure 5.** Some views of the three CEDEX flumes: (**a**) small, (**b**) medium, and (**c**) large.

*2.6. Dimensionless Variables*

Using the height of the physical models (*H*) and the acceleration of gravity (*g*) as the basic variables, and by applying the *Buckingham* Π theorem we can obtain the dimensionless unit discharge ($q^*$) expressed by Equation (5) and the dimensionless equivalent Darcy's coefficient of permeability ($k_{eq}^*$) expressed by Equation (6).

$$q^* = \frac{q}{\sqrt{g \cdot H^3}} \tag{5}$$

$$k_{eq}^* = \frac{k_{eq}}{\sqrt{g \cdot H}} \tag{6}$$

Even though Darcy's law is not applicable in coarse materials such as those used in this study, the equipotential lines at the toe of a rockfill shoulder with linear and nonlinear models are nearly vertical [51]. Assuming the maximum hydraulic gradient at the toe of the rockfill dam as being $i_{max} = 1/Z_{dss}$, then parameters *a* and *b* of the nonlinear resistance law (Equation (4)) can be converted into a single equivalent Darcy's coefficient of permeability ($k_{eq}$) using Equation (7) [10]. The velocity $v_{max}$ is that occurring for the maximum gradient $i_{max}$ at the toe of the dam.

$$k_{eq} = \frac{v_{max}}{i_{max}} = \frac{Z_{dss} \cdot \left(-a + \sqrt{a^2 + 4b/Z_{dss}}\right)}{2b} \tag{7}$$

To compare physical models with different geometries and dimensions, the horizontal lengths were also converted to non-dimensional ($x^*$) using Equation (8). This dimensionless variable ranges from zero to one, from the downstream edge of the crest to the toe of the dam.

$$x^* = \frac{x}{Z_{dss} \cdot H} \tag{8}$$

*2.7. Statistical Analyses*

The statistical analyses presented in this paper were all performed with the statistical software R (version 4.0.3). The regression models were obtained with the 'lm' function (R 'stats' Package), the hypotheses contrasts were performed with the 't.test' function (R 'stats' Package), and the power analyses with the 'cohen.d' function (Package 'effsize') to calculate the effect size statistics and the 'pwr.2p2n.test' function (Package 'pwr) to compute the power of the test. The contrast hypotheses between the two groups of samples were performed assuming the two variances as equal.

## 3. Results

*3.1. Failure Initiation and Progress*

The failure progress was observed to be the same for all tests. These observations were in line with those obtained by other authors [3,19,23,28] for highly permeable materials.

Given the test procedure and regardless of the type of flow (overtopping or throughflow), material gradings, and dam geometry, failure always initiated at the toe of the dam for a unit discharge that must overcome a given threshold ($q_{fi}$). Once this threshold was overcome, failure progressed upwards until a new equilibrium state was achieved for a given constant inflow discharge. The failure progress or 'failure paths' obtained during the specific campaign for analyzing the variability of the results are shown in Figure 6.

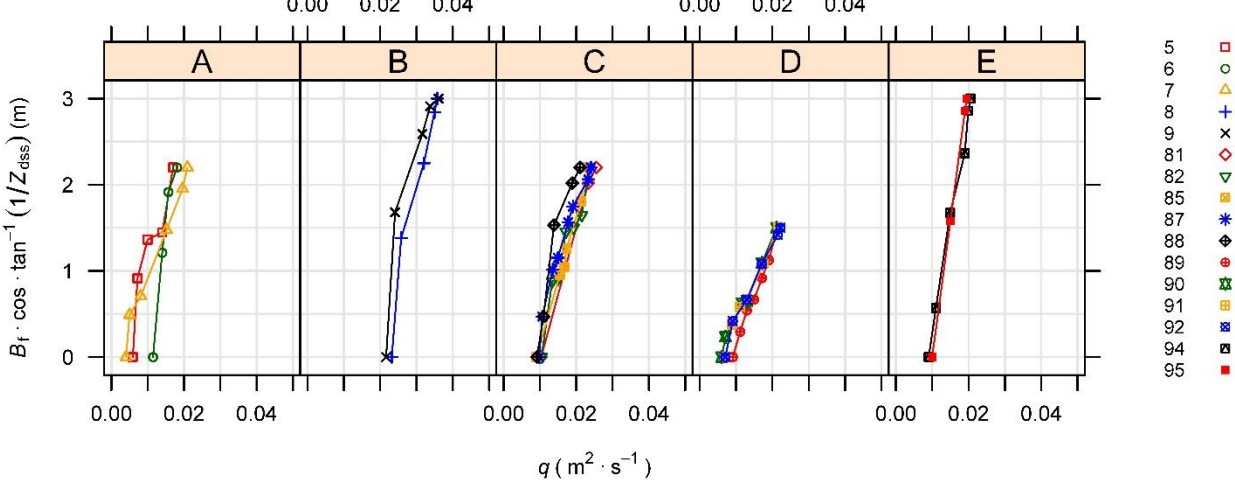

**Figure 6.** Failure paths obtained with the set of tests performed to assess the variability of the results. Each series represents a physical model with several discharge steps.

### 3.2. Failure Discharge

The failure discharge was assumed to be that by which damages inflicted to the downstream shoulder reached the crests of the physical models. For practical reasons, in general, it was defined as the average value between the last discharge step in which damages did not reach the crest ($Q_{f,pre}$) and the first in which damages surpassed it ($Q_{f,pos}$). Table A1 in Appendix A summarizes the average failure discharges ($Q_f = Q_{f,ave}$) obtained for every physical model. Although difficult to compare with other studies from the state of the art (different materials, criteria, etc.), these results are roughly in line with the results of other authors. Test nº 84 ($H = 0.5$ m, $W = 0.4$ m, $Z_{dss} = 1.5$, D50 = 17.3 mm) resulted in a $q_f = 0.0168$ m$^2$ s$^{-1}$, in the same order of magnitude as similar tests performed by Franca and Almeida [28] ($H = 0.5$ m, $W = 2$ m, $Z_{dss} = 1.5$, D50 = 18.9 mm), which obtained a value of $q_f = 0.0138 \pm 0.009$ m$^2$ s$^{-1}$.

### 3.3. Hydraulic Pressures

The hydraulic pressures at the base of the dam were measured for the majority of the tests and all discharge steps. Nonetheless, for simplicity, here, only those pressures relevant for the analysis of the results are presented. Table A2 in Appendix A summarizes the average hydraulic pressures at the bases of the physical models nº 85, 87, 90, 94, and 95 for a given discharge step in the early stages of each test, and Table A3 the average hydraulic pressures from tests nº 87, 94, and 95 for a transversal section roughly located at $x^* = 0.3$ (i.e., 30% of the base length from the crest). The average values were obtained using the records of each piezometer in the same transversal row.

### 3.4. Failure Mechanisms

The laboratory tests allowed the identification of two dominant failure mechanisms: particle dragging (PD) and mass sliding or slumping (MS). Slumping occurs predominantly in embankments with steep slopes. In these cases, failure of the downstream slope affected the entire width of the physical model (Figure 7a). Particle dragging is the predominant failure mechanism in embankments with gentle slopes. In these cases, we observed the formation of one or more erosion channels whose final width was smaller than the total

width of the physical model (Figure 7b). Figure 7 shows digital elevation models (DEM) resulting from the difference between the original undeformed and the failed embankment for different discharge steps (from top to bottom, the discharge is increasing). Light colors represent eroded areas, while dark colors represent areas of deposition.

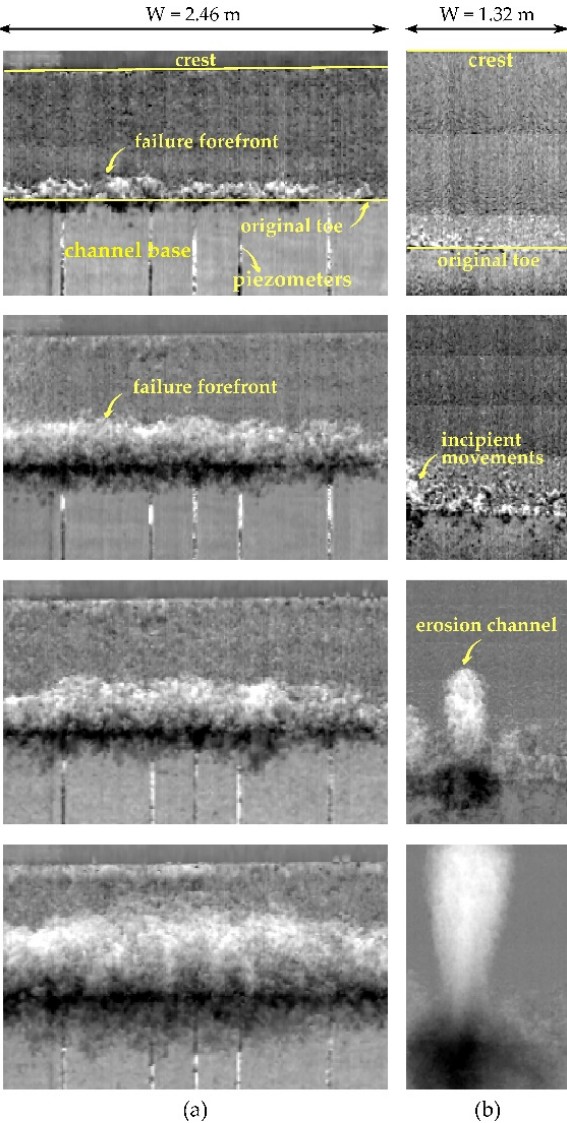

(a)                    (b)

**Figure 7.** Two dominant mechanisms of failure: (**a**) mass sliding and (**b**) particle dragging.

## 4. Discussion

### 4.1. The Mechanics of Failure

Particle dragging is all about the individual stability of each particle when subjected to throughflow forces and gradients as well as skimming flow over the shoulder surface. Once a given discharge threshold is overcome, the motion of a single particle (not a group of particles) is observed, changing the stability conditions of the adjacent particles. If this is a 'key' particle, a type of chain reaction will be triggered, leading the adjacent particles to also move downstream, forming an erosion channel. If the particle is not categorized as key, the adjacent particles will remain stable and in place. In the early stages of failure, this mechanism could lead to the formation of several incipient erosion channels along the entire toe of the dam, but in a given moment, only a few will prevail and grow upstream. Here, the seepage conditions could change significantly, leading to the concentration of flow in the prevailing channels and forcing failure to progress through them. Eventually, only one will prevail, completing the failure process of the downstream shoulder. These

erosion channels are hourglass-shaped (Figure 7b) with steeper walls than the original slope and widths smaller than the total flume width. If key particles are displaced, these and the corresponding adjacent particles will fall radially into the erosion channel, making it progress upwards as well as laterally. Nevertheless, if the displaced particles are not key, this will deepen the erosion channel. The deepening of the channel also occurs naturally as the erosion channel progresses upstream. This phenomenon could eventually lead to slumping. In this case, slumping is not the main failure mechanism but a consequence of particle dragging.

Mass sliding or slumping is related to a problem of global instability affecting a certain mass of material and associated with pore water pressures inside the dam. The sliding mechanism is difficult to detect because the sliding surfaces are usually shallow and quasi-parallel to the slope. A way of detecting sliding is by observing the simultaneous movement (not consecutive) of a group of particles. The physical models where this failure mechanism is dominant are not immune to particle dragging. In these cases, if key particles are displaced as a consequence of sliding, this would also trigger a chain reaction like that described previously, leading the failure forefront, which in these cases usually covers the entire width of the model (Figure 7a), to progress upwards.

### 4.2. The Scale Effect

To compare both 'small-scale' physical models and 'prototypes', the Froude similitude was applied to $q_f$. From this similitude theory, it follows that the unit flow discharge scale factor is $s_q = s_L^{3/2}$. So, comparing both scaled and prototype results of $q_f$ (Table 3), we obtain errors of 2.47%, 2.43%, and 2.75% for models with $Z_{dss} = 1.5$, 2.5, and 3.5, respectively. These errors represent the difference between both results relative to the 'prototype' value. The mean value for the errors is then 2.55% (mean) $\pm 0.14\%$ (standard deviation).

**Table 3.** Summary of the specific campaign to assess the scale effect.

| 'Prototype' Tests | | | 'Small-Scale' Models | | | |
|---|---|---|---|---|---|---|
| **Test** | $Z_{dss}$ | $q_f$ (m$^2$s$^{-1}$) | **Test** | $Z_{dss}$ | $q_f$ (m$^2$ s$^{-1}$) | $q_{f,scaled}$(m$^2$ s$^{-1}$) |
| 108 | 1.5 | 0.032749 | 109 | 1.5 | 0.005125 | 0.033558 |
| 110 | 2.5 | 0.042304 | 111 | 2.5 | 0.006618 | 0.043334 |
| 112 | 3.5 | 0.042505 | 130 | 3.5 | 0.006313 | 0.041337 |

Even though it is a simplistic approach, the Froude similitude to scale between 'physical model' and 'prototype' by scaling D50 seems to give good results for this scale factor. Given that it is the phreatic surface elevation flowing through the downstream shoulder and the first emergence point (intersection between the phreatic surface and the downstream slope) that govern failure and define how far it will progress, then, an alternative approach could be scaling the 'unit discharge–permeability' ratio, as this would more accurately scale the water table elevation. By scaling D50, we are indirectly scaling the permeability, albeit by a factor that we do not know and that should be, in theory, $s_L^{1/2}$, the scale factor for a velocity. Given the small errors between the prototype and scaled $q_f$, we can conclude that for these uniform gravels, the 'unit discharge–permeability' ratio is being somehow properly scaled by scaling D50 with the limits of size existing in this study.

Another problem associated with scaling D50 is that we are managing gravels with different repose angles in the 'prototypes' and 'small-scale' physical models. A failed shoulder profile presents three different slopes, (i) the original slope not yet affected by the failure, (ii) a zone over the phreatic surface with the dry repose angle, and finally (iii) a zone below the phreatic surface and flow with a submerged repose angle. Because it is the slope over the phreatic surface with the dry repose angle that is defines whether failure reaches the crest of the dam or not, then complete failure of the downstream shoulder depends greatly on the gravel that is being tested. Because the repose angle flattens by reducing the size of the granular materials, that could imply that physical models tested

with smaller materials could reach complete failure earlier than if they were tested with coarser materials.

### 4.3. Repeatability

The set of tests dedicated to analyzing repeatability was used to quantify the variability of the results. First, the hydraulic pressures were compared between tests of the same group. Here, we preferred to compare pressures for an early stage of failure, preferably before any major damage was observed to the downstream shoulder because, as failure progresses, the flow net also changes, especially for those cases where particle dragging is the dominant failure mechanism. In these cases, characterized by the formation of one or more erosion channels acting as boreholes or wells, the flow net could change significantly along the width of the physical models suffering a dropdown where the erosion channels are located. As can be observed in Figure 8, the differences between tests n⁰ 85 and 87 (Group C) and n⁰ 94 and 95 (Group E) are negligible. This figure presents the hydraulic pressures measured for roughly the same discharge, 0.0051 and 0.0067 $m^2 \, s^{-1}$ in tests 85 and 87, respectively, and of 0.0071 and 0.0072 $m^2 \, s^{-1}$ in tests 94 and 95. A maximum difference of 0.021 m was observed between tests 85 and 87 for the most upstream measuring section.

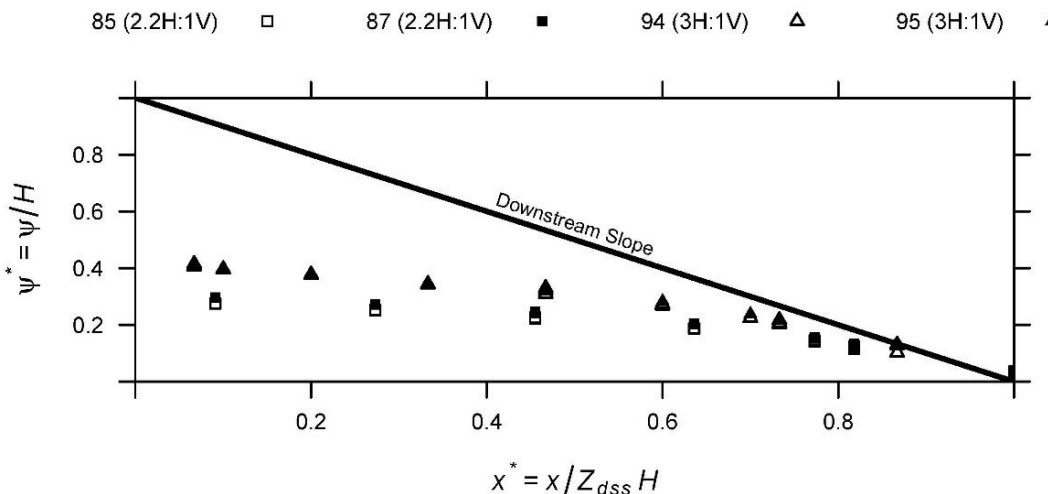

**Figure 8.** Hydraulic pressures measured for roughly the same discharge in tests n⁰ 85 and 87 (Group C) and n⁰ 94 and 95 (Group E). These pressures correspond to discharges of 0.0051 $m^3 \, s^{-1}$ and 0.0067 $m^3 \, s^{-1}$ in tests 85 and 87, respectively, and of 0.0071 $m^3 \, s^{-1}$ and 0.0072 $m^3 \, s^{-1}$ in tests 94 and 95.

By observing the failure paths plotted in Figure 6, it can also be concluded that, in general, tests performed under the same group of tests presented the same trajectories. When failure was complete, groups A, B, C, D, and E resulted, respectively, in unit failure discharges around 0.0187 $m^2 \, s^{-1}$ (mean) $\pm$ 0.0020 $m^2 \, s^{-1}$ (one standard deviation), 0.0357 $\pm$ 0.0005, 0.0234 $\pm$ 0.0016, 0.0216 $\pm$ 0.0005, and 0.02 $\pm$ 0.0007. The ratios of the standard deviation to the mean, i.e., the coefficients of variation (CV), were, also respectively, 10.9%, 1.4%, 6.9%, 2.1%, and 3.7%, being the mean value of CV = 5.0% $\pm$ 3.9%. It can be stated that the results varied within reasonable ranges given the great amount of uncertainty associated with this kind of test.

### 4.4. The Effect of the Downstream Slope

The downstream slope was observed to greatly affect the failure mechanism. The chart presented in Figure 9 was plotted using only the results for the physical models wider than 1 m to avoid having the flume walls affecting the correct development of the erosion channels if that was the case. Also, we only used those models where one of the two mechanisms was dominant at the final stages of failure. In summary, a total of forty-one

physical models were used. The transition between mechanisms occurs for a range of $Z_{dss}$ varying roughly between 2.0 and 2.5. If we focus on the tests with $Z_{dss} = 2.2$, it can be observed that eight out of the nine models failed by particle dragging and resulted in the formation of erosion channels. So, for practical purposes, steep slopes were defined to be those steeper than 2.2 and gentle slopes as those smoother than this value. This could be a simplistic way of categorizing between slopes given that the repose angle may most certainly affect it. Nonetheless, the small variation in these angles within the rockfill materials must be taken into account.

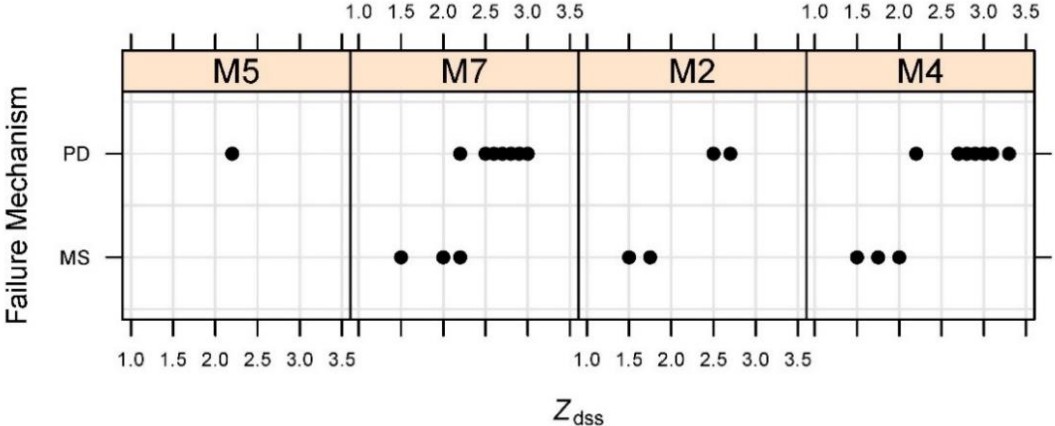

**Figure 9.** Relationship between the failure mechanisms and the downstream slope ($Z_{dss}$). The acronyms PD and MS refer to 'particle dragging' and 'mass sliding', respectively.

The effect of the slope was also noticed in the hydraulic pressures, which increased as the slope was gentler for the same unit flow. This fact is shown in Figure 10, which presents the hydraulic pressures measured for the first discharge step in tests nº 87, 90, 94, and 95. All of these tests were of the type CPM/NIE, i.e., complete configurations without impervious element and, thus, throughflow. For relative distances of $x^* = 0.2$, 0.5, and 0.7, an embankment with a slope $Z_{dss} = 3.0$ resulted, respectively, in hydraulic pressures 47.3%, 59.6%, and 78.9% higher than those resulting from an embankment with a slope $Z_{dss} = 1.5$.

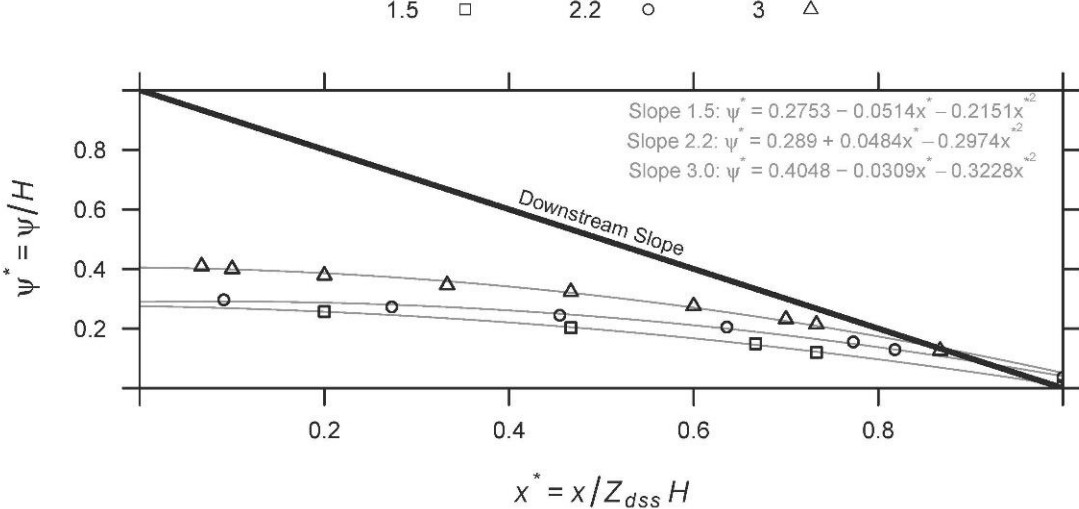

**Figure 10.** Hydraulic pressures measured for roughly the same discharge (0.007 m$^3$ s$^{-1}$) in tests nº 87 [$Z_{dss} = 2.2$], nº 90 [$Z_{dss} = 1.5$], and nº 94 and 95 [$Z_{dss} = 3.0$]. The plotted pressures for $Z_{dss} = 3.0$ were averaged between both tests (Figure 8). Although pressures measured in test nº 85 [$Z_{dss} = 2.2$] were similar to that measured in test nº 87 (Figure 8), in this figure it was decided not to average them because in test nº 85 they correspond to a discharge of 0.005 m$^3$ s$^{-1}$.

Figure 11 presents the evolution of the hydraulic pressure measured during tests nº 87, 94, and 95 in a single section of the physical models, roughly located at $x^* = 0.3$ (30% of the base length from the crest).

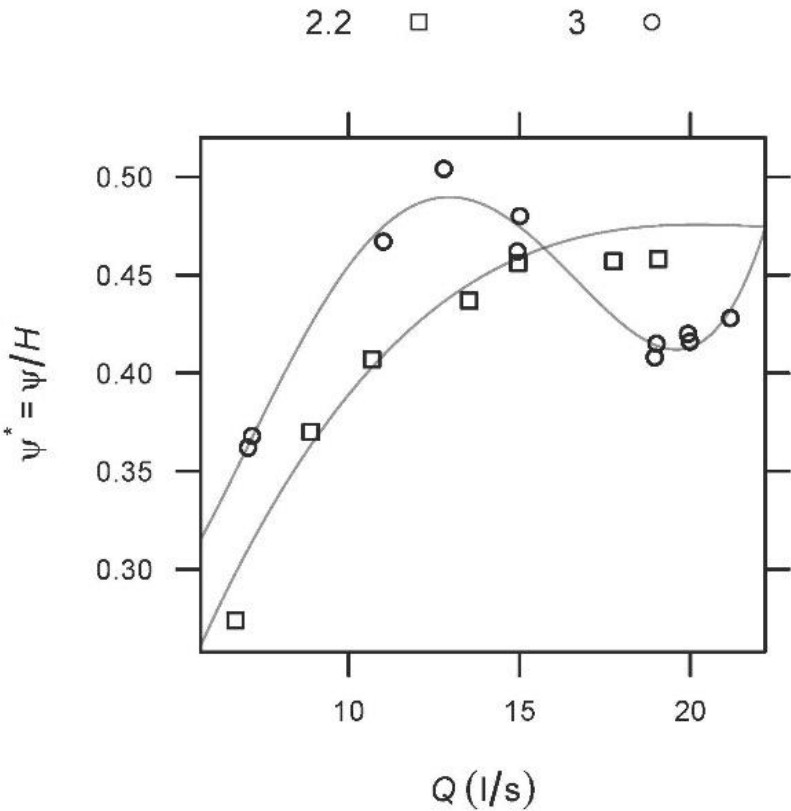

**Figure 11.** Hydraulic pressures measured in a single section of the physical models located at $x^* = 0.3$ (30% of the base length from the crest) in tests nº 87 [$Z_{dss} = 2.2$] and nº 94 and 95 [$Z_{dss} = 3.0$]. The plotted pressures for $Z_{dss} = 3.0$ were averaged between both tests (Figure 8).

Here, it can also be seen that, for a given unit flow, the gentler the slope, the higher the hydraulic pressures, at least in the early stages of the tests. In a given moment, i.e., for a given discharge step, pressures measured in the physical models with $Z_{dss} = 3.0$ suffered a sudden decrease that matched the formation of an erosion channel, as can be observed in Figure 12. It must be noted that this pressure dropdown was not observed in test nº 87 ($Z_{dss} = 2.2$) even though the failure mechanism of this test was the same as that of tests nº 94 and 95 ($Z_{dss} = 3.0$), as can be observed in Figure 13. This observation denotes that the failure mechanism is not enough to explain the pressure dropdown and that other variable/s should also be considered—for example, the geometry and dimensions of the erosion channel.

The repeatability campaign groups C ($Z_{dss} = 2.2$), D ($Z_{dss} = 1.5$), and E ($Z_{dss} = 3.0$) were compared with each other to assess the possible effect of this variable on $q_f$. From a physical point of view, and considering only the range of slopes for which the dominant failure mechanism is mass sliding or slumping, it could be expected that the steeper slopes within this range resist higher flow discharges than gentler slopes as a result of the higher hydraulic gradients that lead to lower phreatic surfaces and lower pressures. This hypothesis was confirmed in Figure 10. If we now expand the range of slopes and compare those equal to 1.5 and 3.0 through a one-sided test, we obtain that steep slopes completely fail for higher flow discharges than gentle slopes for a $p$-value $= 0.086$ and a power of 64.9%. We could accept the alternative hypothesis (steep slopes resist higher flow discharges) for a 0.1 significance level ($\alpha$), thus having a 10% and 35.1% chance of committing Type I and II errors, respectively. When contrasting the slopes 2.2 and 3.0

also through a one-sided test, we obtain a $p$-value $= 0.035$ and a power of 74.0%. We could accept the alternative hypothesis even for a lower significance level, $\alpha = 0.05$, thus having a 5% and 26.0% chance of committing Type I and II errors, respectively. Therefore, steeper slopes seem to resist higher unit discharges than gentler slopes, a trend that is also observed in Figure 14. Nonetheless, it must be noted that this tendency was not observed, for example, during the scale effect campaign, summarized in Table 3, nor when comparing slopes 1.5 and 2.2 also through a one-sided test, where we obtained a $p$-value $= 0.955$ and a power of 0.06% (for $\alpha = 0.1$). These are contradictory results that should be examined more deeply in future investigations. They could be related more to testing variability (tests performed throughout different R&D projects, different laboratories, measurement techniques, discharge steps, etc.) rather than the variability of the results that we already saw to be small when repeating the same test in the same conditions.

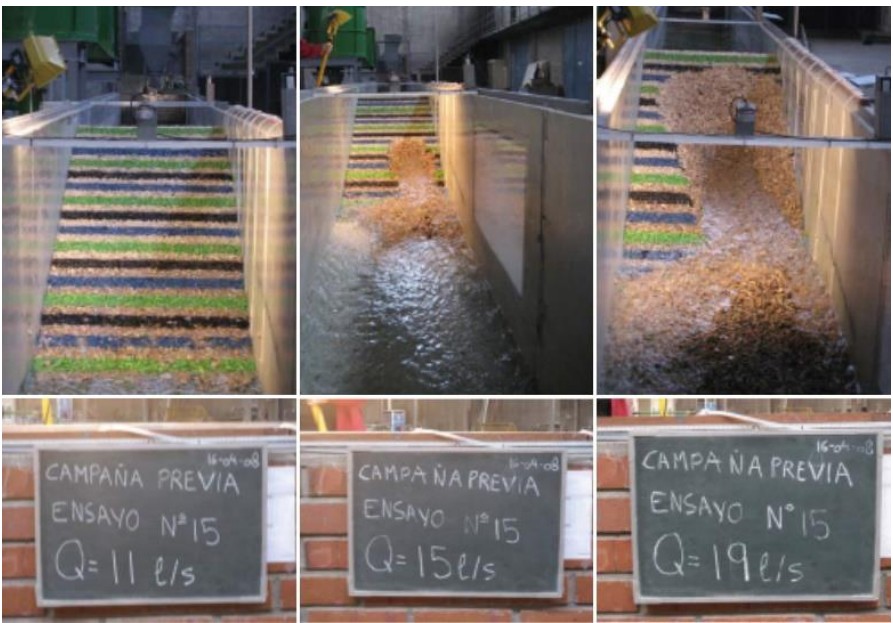

**Figure 12.** Images of test nº 95 [$Z_{dss} = 3.0$] before and during the pressure dropdown (Figure 11).

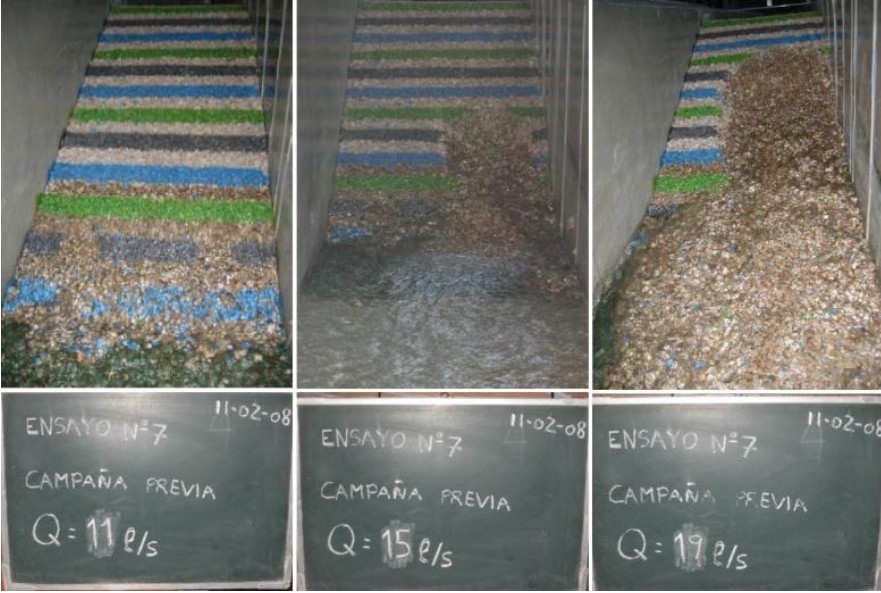

**Figure 13.** Images of test nº 87 [$Z_{dss} = 2.2$].

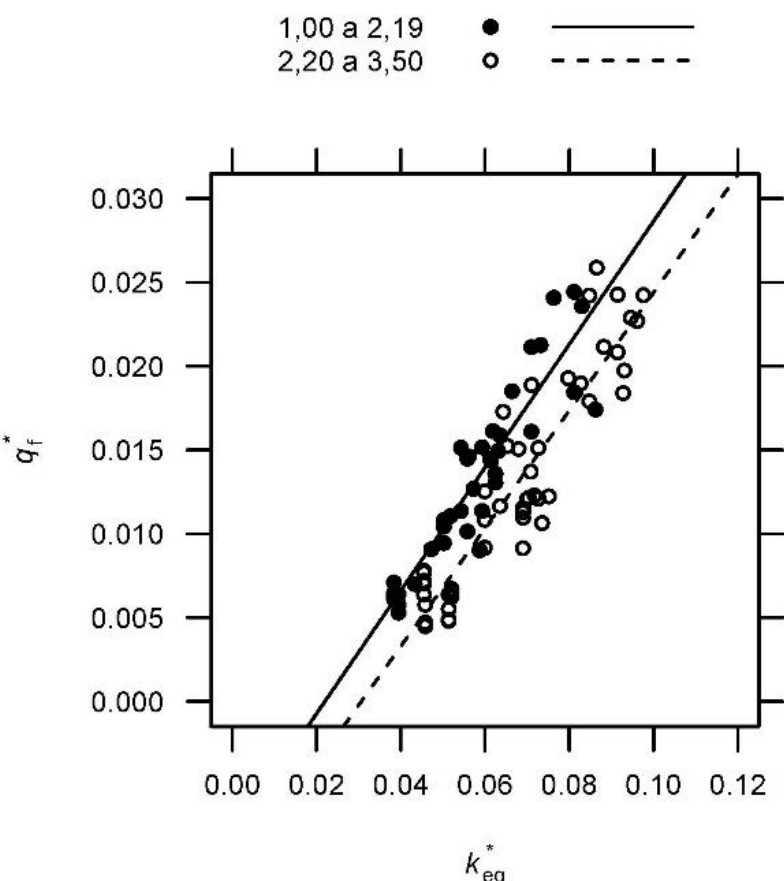

**Figure 14.** Variation of $q_f^*$ with $k_{eq}^*$ differentiating by the downstream slope steepness.

One last observation must be taken into consideration. Here, we would like to emphasize that we were dealing with slopes that were unstable in throughflow conditions, i.e., we were dealing with physical models that failed. So, if a gentle slope, stable to slumping, is not subjected to flow discharges capable of dragging its particles, then this embankment will remain stable. On the other hand, the same dam constructed with a steeper slope unstable to slumping would fail, so in this case, gentler slopes would be more resistant.

*4.5. Other Effects*

There are clear differences between the physical processes related to overtopping and throughflow. Nonetheless, characteristics of throughflow such as pressures or the phreatic surface elevation (including the first emergence point) are not significantly different between both types of flow [10]. So, because it is the water level inside the rockfill dam and the position of the first emergence point that governs the failure of the downstream shoulder by determining how far it will progress, and since these do not change significantly when changing the type of flow, we would expect to obtain a similar failure discharge $q_f$ in each case, for example, CPM/UF vs. CPM/NIE or PPM/CC vs. PPM/NIE (Figure 1). Results of the statistical analysis performed for tests nº 140, 141, 142 (CPM/NIE) and nº 143 and 146 (CPM/UF) support this idea. The *p*-value = 0.027, resulting from a two-tailed test, allows us to reject the alternative hypothesis for $\alpha = 0.01$ and claim that the type of flow does not affect $q_f$. The statistical power of this analysis is about 93%, so we have a small probability of also committing the Type II statistical error. In other words, we have a 7% chance of rejecting the alternative hypothesis when it should be accepted. Besides this comparison, Figure 15 shows the relation between $k_{eq}^*$ and $q_f^*$ differentiating between tests performed with overtopping and throughflow. Differences between both scatter plots are negligible, especially in the lower part of the chart.

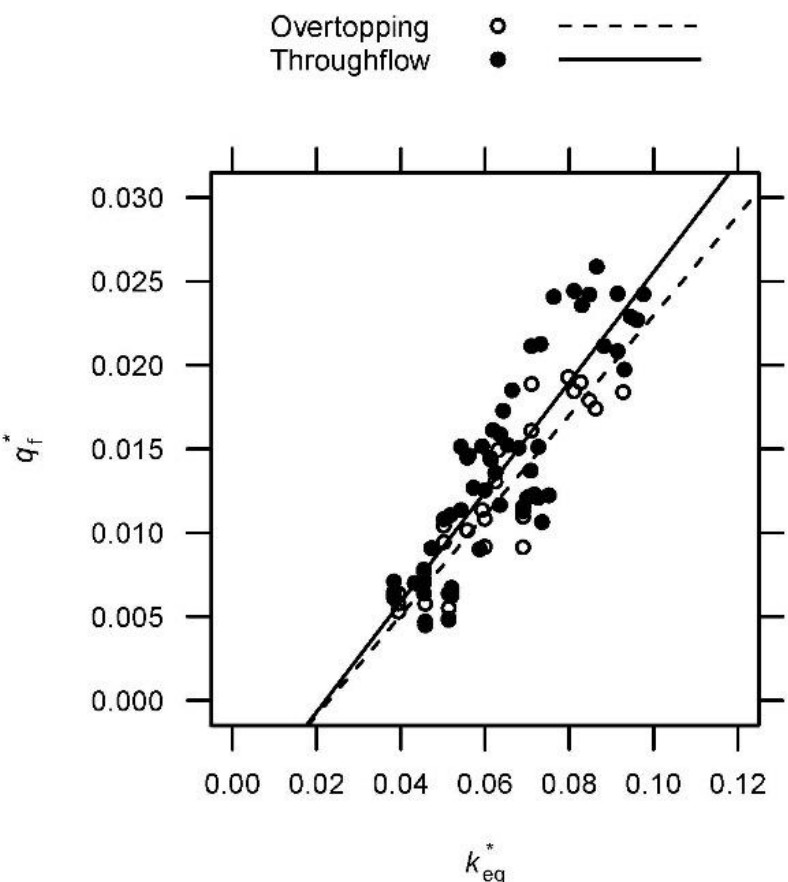

**Figure 15.** Variation of $q_f^*$ with $k_{eq}^*$ differentiating by the type of flow.

Those physical models where the differentiating variable was the type of impervious element (central core or upstream face) were also compared. In this case, we could somehow expect to observe some differences in the value of $q_f$. The process through which water infiltrates was the same, overtopping, but the flow paths inside the body of the dam were longer for dams with an upstream face, which did not necessarily imply a difference in the position of the first emergence point. Results from the statistical analysis performed on tests nº 22 and 123 (PPM/CC) and nº 124 and 128 (CPM/UF) indicate that the type of impervious element does not have a significant impact on the value of $q_f$. The $p$-value = 0.124 allows us to reject the alternative hypothesis for an $\alpha = 0.05$. The statistical power is about 73%, so here we have a 27% chance of rejecting the alternative hypothesis when it should be accepted. Crossing these results with Figure 16 makes us confident to state, for now, that the type of impervious element does not affect $q_f$. Figure 17 presents the images of both tests and discharge steps used in the construction of the Figure 16 plot. Besides this comparison, Figure 18 shows the relation between $k_{eq}^*$ and $q_f^*$ differentiating between tests performed with upstream face (UF) and central core (CC). Here, we also cannot see any clear separation between both scatter plots.

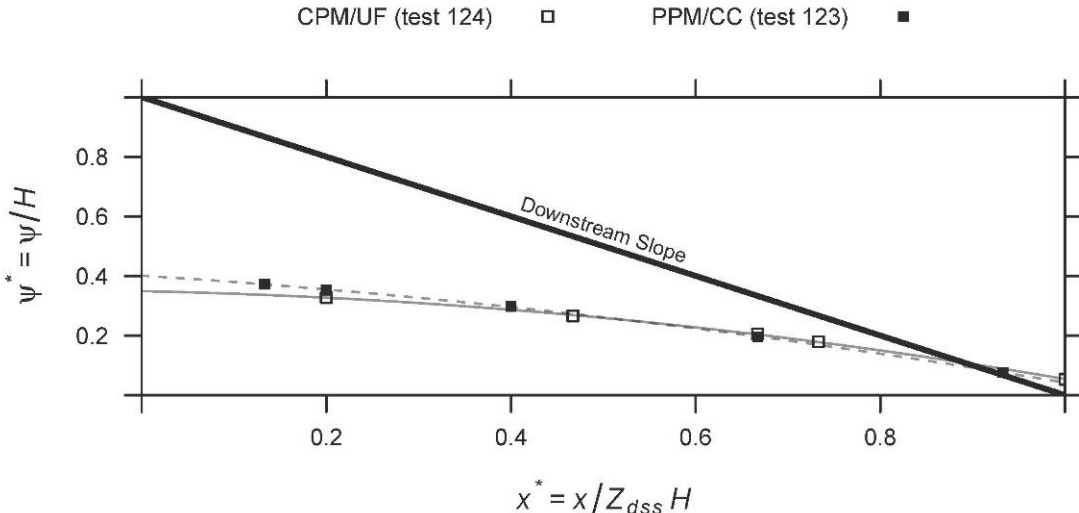

**Figure 16.** Hydraulic pressures measured for a discharge of 0.0091 and 0.0090 m$^3$ s$^{-1}$ in tests nº 123 and 124, respectively.

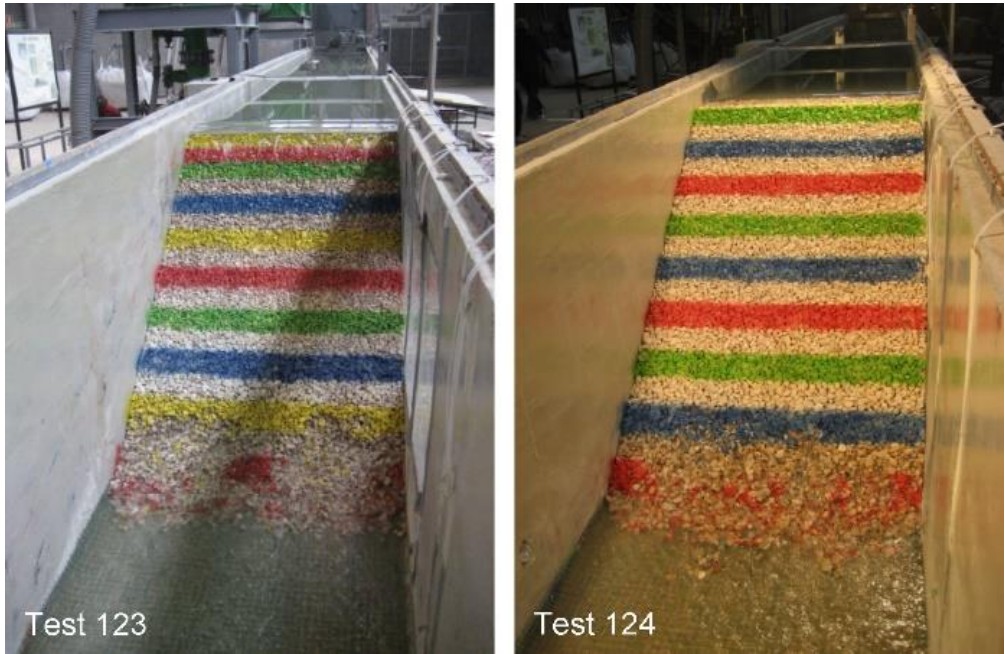

**Figure 17.** Failure progress in tests nº 123 and 124 for a discharge of roughly 0.0090 m$^3$ s$^{-1}$.

### 4.6. Research Scope and Limitations

The inferential analysis seeks patterns from a population through samples of it. In this kind of analysis, the aphorism 'more is better' referring to the sample size is true because samples with a higher number of observations imply smaller confidence intervals and more reliable conclusions. But the truth is that researchers are most of the time far from this ideal goal working even with extremely small samples. In these cases, statistical power analysis assumes an important role by including the probability of committing Type II errors besides the traditional assessment of the Type I error [52]. Alongside size sampling, this study contains a series of other limitations, including, for example, the size of the physical models that could lead to scale effects, the use of uniform limestone materials, preventing a thorough analysis of the effect of other material parameters, and the small number of tests repeated in the same conditions.

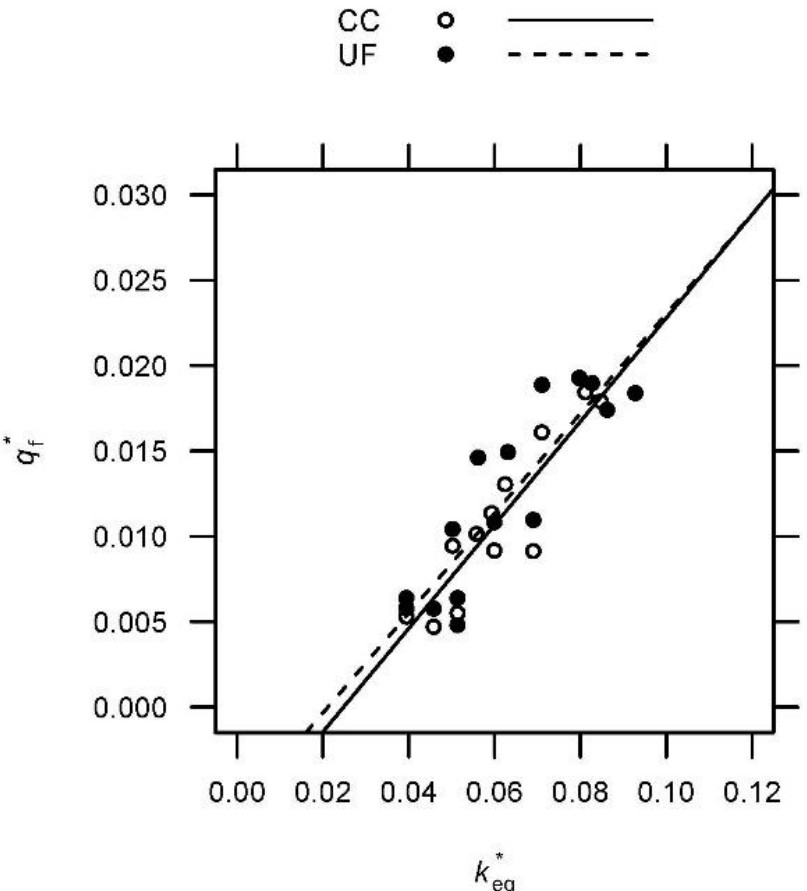

**Figure 18.** Variation of $q_f^*$ with $k_{eq}^*$ differentiating by the impervious element (Central Core and Upstream Face).

*4.7. A Regression Model for the Failure Discharge*

Taking into account the discussion of the results presented in the previous sections and the scope and limitations of the research, we propose a regression model that can be used to estimate the failure discharge of a rockfill downstream shoulder ($q_f$), i.e., that overtopping or throughflow discharge with a failure degree that reaches the crest of the dam. This model depends on the equivalent Darcy's coefficient of permeability ($k_{eq}$), expressed by Equation (7), on the downstream slope ($Z_{dss}$) and height ($H$) of the dam, and on the acceleration of gravity ($g$). To be precise, two regression models were calibrated, one for 'steep' and one for 'gentle' slopes, with the critical slope $Z_{dss} = 2.2$ for the granular materials used in this research and an angle of repose around $41°$. Both models pass through the origin since this is a non-tested data point, i.e., in the limit, a dam with $k_{eq}^* = 0$ should need no overtopping to fail. Equations (9) and (10) should be used for gentle and steep slopes, respectively. If no distinction is desired between these two categories, then Equation (11) should be used. The first two equations both have a coefficient of determination $R^2 = 0.97$, while the third uses 0.95. It should be noted that tests 114 and 129 were excluded from Equation (9) because these were outliers with Cook's distances of roughly 1 and 0.5, respectively, and that these calibrations included only those tests where the permeability of the materials was obtained. Equations (9) and (10) can be observed fitted to data in Figure 19.

$$q_{f,gentle} = 2.659 \cdot \sqrt{\frac{H}{g}} \cdot k_{eq}^{2} \qquad (9)$$

$$q_{f,steep} = 3.610 \cdot \sqrt{\frac{H}{g}} \cdot k_{eq}^{2} \qquad (10)$$

$$q_f = 2.937 \cdot \sqrt{\frac{H}{g}} \cdot k_{eq}^2 \tag{11}$$

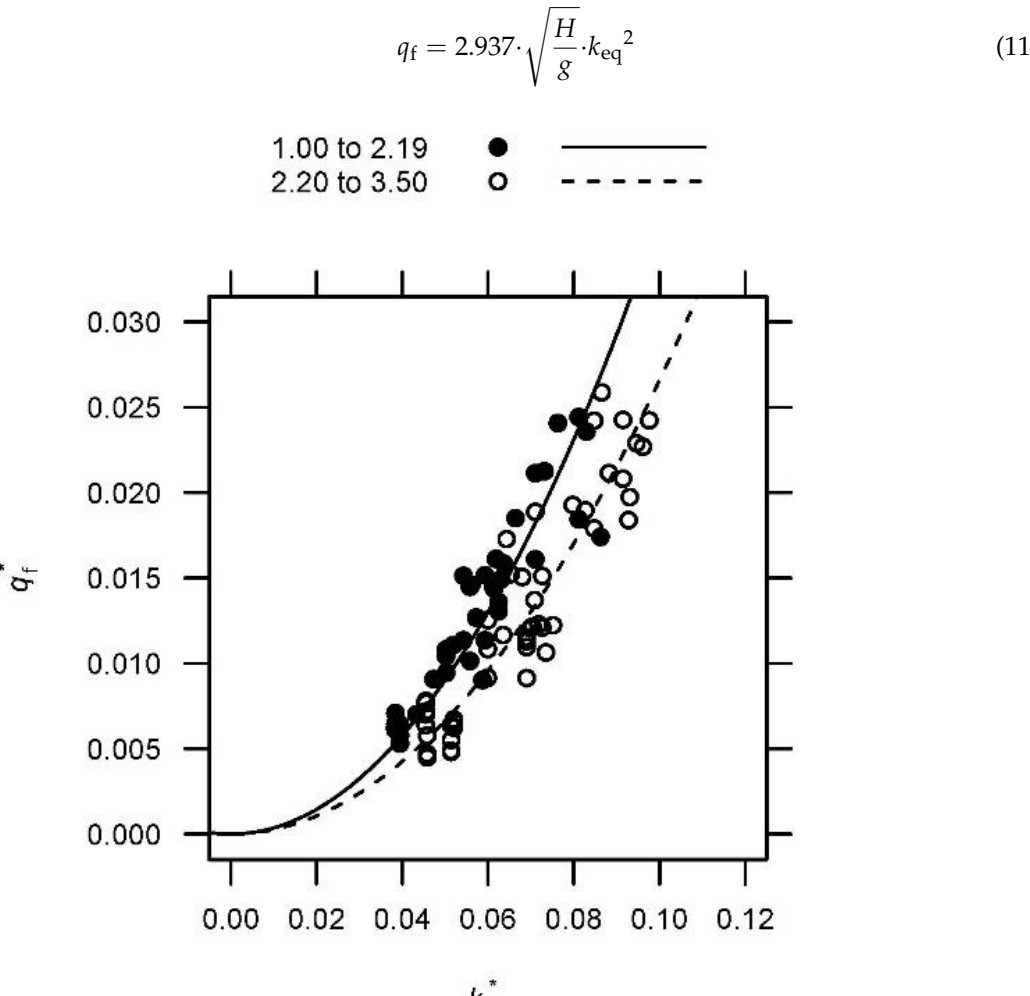

**Figure 19.** Regression models proposed to estimate the failure discharge ($q_f$) for dams with steep slopes ($Z_{dss} < 2.2$) and gentle slopes ($Z_{dss} > 2.2$).

## 5. Conclusions

This study allowed the identification of two different failure mechanisms for highly permeable rockfills subjected to overflowing: slumping and particle dragging. The occurrence of one or the other is heavily dependent on the slope of the embankment and the size of the particles. Slumping is related to a problem of global instability of a certain mass of material and is predominant in embankments with steep slopes ($Z_{dss} < 2.2$). In these cases, failure of the downstream slope affected the entire width of the physical models. On the other hand, when particle dragging is the predominant failure mechanism, it means that we are dealing with slopes that are stable to slumping. In these cases, which are all about the individual stability of each particle when subject to seepage forces and hydraulic gradients, we observed the formation of erosion channels whose width was smaller than the total width of the physical models. The critical slope, i.e., the slope that defines the limit for the occurrence of one mechanism or the other, was identified to be $Z_{dss} = 2.2$ for the characteristics of the materials used in this study.

The downstream slope was also observed to affect both the hydraulic pressures at the base of the physical models and the value of the failure discharge, i.e., the inflow that forces failure to reach the crest of the dam. For a discharge of 0.007 m$^3$ s$^{-1}$, the hydraulic pressures measured in a dam with $Z_{dss} = 3.0$ were, on average, 61.9% higher than in a dam with $Z_{dss} = 1.5$. Regarding failure, steep slopes ($Z_{dss} < 2.2$) were prone to fail for higher discharges than gentle slopes ($Z_{dss} > 2.2$) or, in other words, tended to be, on average, more resistant. Given this observation, two regression models were proposed to estimate

the unit failure discharge ($q_f$) of dams with gentle and steep slopes, Equations (9) and (10), respectively, both functions of the equivalent Darcy's coefficient of permeability ($k_{eq}$), expressed by Equation (7), the downstream slope ($Z_{dss}$), the height of the embankment ($H$), and the acceleration of gravity ($g$). If no distinction is desired to be made between slopes, then $q_f$ should be estimated with Equation (11).

Other factors such as the type of flow (overtopping or throughflow) or the type of impervious element (central core or upstream face) were not observed to affect the hydraulic pressures or the failure discharge.

**Author Contributions:** Conceptualization, R.M.-A., M.Á.T. and R.M.; methodology, M.Á.T. and R.M.; software, R.M.-A.; validation, R.M.-A., M.Á.T.; formal analysis, R.M.-A. and M.Á.T.; investigation, R.M.-A., R.M. and L.B.; resources, M.Á.T. and L.B.; data curation, R.M.-A.; writing—original draft preparation, R.M.-A.; writing—review and editing, M.Á.T. and R.M.; visualization, R.M.-A.; supervision, M.Á.T. and R.M.; project administration, R.M.-A., M.Á.T., R.M. and L.B.; funding acquisition, M.Á.T., R.M. and L.B. All authors have read and agreed to the published version of the manuscript.

**Funding:** This research was funded by the Spanish Ministry of Science and Innovation, grant number BIA2010-21350-C03-03 (Project EDAMS—Rotura del elemento impermeable de presas de materiales sueltos en situación de sobrevertido y análisis de protecciones combinando modelación física e inteligencia artificial), and by the Spanish Ministry of Economy and Competitiveness, grant number BIA2007-68120-C03-02 (Project XPRES—Caracterización de la rotura de las presas de escollera por sobrevertido y desarrollo de criterios para evaluar la seguridad del conjunto presa-área afectada durante una avenida).

**Acknowledgments:** Within the scope of the funding research projects, we would like to thank Ángel Lara, Rafael Cobo, Cristina Lechuga, Isabel Berga, and Hibber Campos for their participation in these projects.

**Conflicts of Interest:** The authors declare no conflict of interest. The funders had no role in the design of the study; in the collection, analyses, or interpretation of data; in the writing of the manuscript, or in the decision to publish the results.

## Nomenclature

The following symbols and acronyms are used in this paper:

| | |
|---|---|
| $a$ | Seepage resistance coefficient of the quadratic flow equation (fundamental units $T \cdot L^{-1}$) |
| $b$ | Seepage resistance coefficient of the quadratic flow equation (fundamental units $T^2 \cdot L^{-2}$) |
| $B_f$ | Maximum advance of failure in relation to the embankment toe, i.e., the distance between the most upward point of the failure forefront and the downstream toe as shown in Figure 2 (fundamental units L) |
| CC | Central core |
| CPM | Complete physical model or complete configuration (both upstream and downstream slopes and crest) |
| CPM/UF | Complete configuration with upstream face |
| CPM/NIE | Complete configuration without impervious element |
| $C_u$ | Coefficient of uniformity, the ratio D60/D10 where D60 and D10 are the sieve sizes through which 60 and 10% of the granular material (dimensionless) passes |
| CV | Coefficient of variation |
| D50 | Sieve size passing 50% of the particles (fundamental units L) |
| DSS | Downstream shoulder |
| $e$ | Void ratio (dimensionless) |
| $g$ | Acceleration of gravity (fundamental units $L \cdot T^{-2}$) |
| $H$ | Height of the dam (fundamental units L) |
| $i$ | Hydraulic gradient (dimensionless) |
| $k_{eq}$ | Equivalent Darcy's coefficient of permeability (fundamental units $L \cdot T^{-1}$) |
| $k_{eq}^*$ | Dimensionless equivalent Darcy's coefficient of permeability (dimensionless) |
| L | Length, fundamental dimension |
| $l_c$ | Width of the embankment crest (fundamental units L) |
| M | Mass, fundamental dimension |

| | |
|---|---|
| $n$ | Porosity (dimensionless) |
| NIE | No impervious element |
| PM | Physical model |
| PPM | Partial physical model (partial configurations) |
| PPM/CC | Partial or complete configuration with central core |
| PPM/NIE | Partial configuration without impervious element |
| $q$ | Unit discharge (fundamental units $L^2 \cdot T^{-1}$) |
| $q^*$ | Unit discharge (dimensionless) |
| $q_f$ | Unit failure discharge, i.e., the discharge in which failure reaches the crest of the embankment (fundamental units $L^2 \cdot T^{-1}$) |
| $q_{fi}$ | First unit discharge producing any visible damage to the downstream slope (fundamental units $L^2 \cdot T^{-1}$) |
| $q_{f,pre}$ | Last unit discharge step in which failure does not reach the crest of the embankment (fundamental units $L^2 \cdot T^{-1}$) |
| $q_{f,pos}$ | First unit discharge step in which failure surpasses the crest of the embankment (fundamental units $L^2 \cdot T^{-1}$) |
| $R^2$ | Coefficient of determination |
| $s_L$ | Scale factor |
| sd | Standard deviation |
| T | Time, fundamental dimension |
| UF | Upstream face |
| USS | Upstream shoulder |
| $W$ | Width of the test flumes (fundamental length L) |
| $x^*$ | Horizontal lengths (dimensionless) |
| $Z_{dss}$ | Slope of the downstream rockfill shoulder (dimensionless) |
| $Z_{uss}$ | Slope of the upstream rockfill shoulder (dimensionless) |
| $\varphi'$ | Internal friction angle of gravels (°) |
| $\gamma$ | Dry unit weight (fundamental units $M \cdot L^{-3}$) |
| $\gamma_s$ | Specific gravity of solid particles (fundamental units $M \cdot L^{-3}$) |
| $\gamma_{sat}$ | Saturated unit weight (fundamental units $M \cdot L^{-3}$) |
| $\gamma_w$ | Density of water (fundamental units $M \cdot L^{-3}$) |
| $\psi$ | Pressure head (fundamental units L) |
| $\psi^*$ | Pressure head (dimensionless) |

## Appendix A

**Table A1.** Summary of the failure discharge ($Q_f$) obtained for every physical model tested at the laboratory. The subscript 'pre' refers to the last discharge in which failure did not reach the crest, and 'pos' refers to the first in which failure surpasses it. The subscript 'ave' refers to the average value of the previous two. NA means 'not available'.

| Test | Laboratory | $H$ (m) | $W$ (m) | $Z_{dss}$ (H:V) | $Z_{uss}$ (H:V) | $l_c$ (m) | IE | Gravel Code | $Q_{f,pre}$ (L·s⁻¹) | $Q_{f,pos}$ (L·s⁻¹) | $Q_f$ (L·s⁻¹) |
|---|---|---|---|---|---|---|---|---|---|---|---|
| 1 | UPM | 0.6 | 2.50 | 2.50 | 1.50 | 0.20 | NIE | M2 | 18.000 | 21.000 | 19.500 |
| 2 | UPM | 0.6 | 2.50 | 1.50 | 1.50 | 0.20 | NIE | M2 | 18.400 | 21.500 | 19.950 |
| 3 | UPM | 0.6 | 2.50 | 1.75 | 1.50 | 0.20 | NIE | M2 | 23.400 | 26.000 | 24.700 |
| 4 | UPM | 0.6 | 2.50 | 2.70 | 1.50 | 0.20 | NIE | M2 | 20.200 | 23.000 | 21.600 |
| 5 | UPM | 1.0 | 2.50 | 2.20 | 1.50 | 0.20 | NIE | M5 | 35.000 | 50.000 | 42.500 |
| 6 | UPM | 1.0 | 2.50 | 2.20 | 1.50 | 0.20 | NIE | M5 | 39.240 | 51.290 | 45.265 |
| 7 | UPM | 1.0 | 2.50 | 2.20 | 1.50 | 0.20 | NIE | M5 | 49.000 | 55.800 | 52.400 |
| 8 | UPM | 1.0 | 2.46 | 3.00 | 1.50 | 0.20 | NIE | M7 | 86.009 | 87.944 | 86.977 |
| 9 | UPM | 1.0 | 2.46 | 3.00 | 1.50 | 0.20 | NIE | M7 | 83.075 | 94.468 | 88.772 |
| 10 | UPM | 1.0 | 2.46 | 2.50 | 1.50 | 0.20 | NIE | M7 | 85.206 | 94.468 | 89.837 |
| 11 | UPM | 1.0 | 2.46 | 2.20 | 1.50 | 0.20 | NIE | M7 | 94.468 | 98.569 | 96.519 |
| 12 | UPM | 1.0 | 2.46 | 1.50 | 1.50 | 0.20 | NIE | M7 | 72.204 | 94.468 | 83.336 |
| 13 | UPM | 1.0 | 2.46 | 2.20 | 1.50 | 0.20 | UF | M7 | 75.512 | 91.380 | 83.446 |

**Table A1.** *Cont.*

| Test | Laboratory | $H$ (m) | $W$ (m) | $Z_{dss}$ (H:V) | $Z_{uss}$ (H:V) | $l_c$ (m) | IE | Gravel Code | $Q_{f,pre}$ (L·s$^{-1}$) | $Q_{f,pos}$ (L·s$^{-1}$) | $Q_f$ (L·s$^{-1}$) |
|------|------------|---------|---------|-----------------|-----------------|-----------|-----|-------------|---------|---------|---------|
| 14 | UPM | 1.0 | 2.46 | 3.00 | 1.50 | 0.20 | UF | M7 | 76.560 | 92.300 | 84.430 |
| 15 | UPM | 1.0 | 2.46 | 1.50 | 1.50 | 0.20 | UF | M7 | 71.040 | 89.610 | 80.325 |
| 16 | UPM | 1.0 | 2.46 | 3.00 | 1.50 | 0.20 | CC | M7 | 63.740 | 77.070 | 70.405 |
| 17 | UPM | 1.0 | 2.46 | 2.20 | 1.50 | 0.20 | CC | M7 | 62.790 | 78.440 | 70.615 |
| 18 | UPM | 1.0 | 2.46 | 1.50 | 1.50 | 0.20 | CC | M7 | 65.180 | 80.400 | 72.790 |
| 20 | UPM | 1.0 | 2.46 | 3.00 | 1.50 | 0.20 | CC | M4 | 39.400 | 45.430 | 42.415 |
| 21 | UPM | 1.0 | 2.46 | 2.20 | 1.50 | 0.20 | CC | M4 | 33.470 | 38.990 | 36.230 |
| 22 | UPM | 1.0 | 2.46 | 1.50 | 1.50 | 0.20 | CC | M4 | 39.560 | 42.280 | 40.920 |
| 23 | UPM | 1.0 | 2.46 | 2.20 | 1.50 | 0.20 | NIE | M4 | 30.630 | 38.470 | 34.550 |
| 24 | UPM | 1.0 | 2.46 | 3.00 | 1.50 | 0.20 | NIE | M4 | 34.830 | 40.420 | 37.625 |
| 25 | UPM | 1.0 | 2.46 | 3.00 | 1.50 | 0.20 | UF | M4 | 36.990 | 37.070 | 37.030 |
| 34 | UPM | 0.5 | 2.46 | 1.50 | NA | NA | CC | M4 | 26.272 | 28.969 | 27.621 |
| 35 | UPM | 0.5 | 2.46 | 1.75 | NA | NA | CC | M4 | 30.167 | 31.706 | 30.937 |
| 36 | UPM | 0.5 | 2.46 | 2.00 | NA | NA | CC | M4 | 33.752 | 37.335 | 35.544 |
| 38 | UPM | 0.5 | 2.46 | 1.50 | NA | NA | CC | M7 | 41.292 | 46.392 | 43.842 |
| 40 | UPM | 0.5 | 2.46 | 2.00 | NA | NA | CC | M7 | 46.365 | 54.097 | 50.231 |
| 41 | UPM | 0.5 | 2.46 | 2.20 | NA | NA | CC | M7 | 46.475 | 51.055 | 48.765 |
| 42 | UPM | 0.5 | 0.60 | 1.00 | NA | NA | NIE | M4 | 5.054 | 7.013 | 6.034 |
| 43 | UPM | 0.5 | 0.60 | 1.50 | NA | NA | NIE | M4 | 9.019 | 10.230 | 9.625 |
| 44 | UPM | 0.5 | 0.60 | 1.75 | NA | NA | NIE | M4 | 8.973 | 11.171 | 10.072 |
| 45 | UPM | 0.5 | 0.60 | 2.00 | NA | NA | NIE | M4 | 9.025 | 9.025 | 9.025 |
| 46 | UPM | 0.5 | 0.60 | 2.25 | NA | NA | NIE | M4 | 8.959 | 11.283 | 10.121 |
| 47 | UPM | 0.5 | 0.60 | 2.50 | NA | NA | NIE | M4 | 9.024 | 10.981 | 10.003 |
| 48 | UPM | 0.5 | 0.60 | 2.75 | NA | NA | NIE | M4 | 7.035 | 9.112 | 8.074 |
| 49 | UPM | 0.5 | 0.60 | 3.00 | NA | NA | NIE | M4 | 9.050 | 11.044 | 10.047 |
| 50 | UPM | 0.5 | 0.60 | 1.00 | NA | NA | NIE | M7 | 4.855 | 7.130 | 5.993 |
| 51 | UPM | 0.5 | 0.60 | 1.50 | NA | NA | NIE | M7 | 13.037 | 15.053 | 14.045 |
| 52 | UPM | 0.5 | 0.60 | 1.75 | NA | NA | NIE | M7 | 14.842 | 17.151 | 15.997 |
| 53 | UPM | 0.5 | 0.60 | 2.00 | NA | NA | NIE | M7 | 15.120 | 17.337 | 16.229 |
| 54 | UPM | 0.5 | 0.60 | 2.20 | NA | NA | NIE | M7 | 15.014 | 17.164 | 16.089 |
| 55 | UPM | 0.5 | 0.60 | 2.40 | NA | NA | NIE | M7 | 13.024 | 15.074 | 14.049 |
| 57 | UPM | 0.5 | 0.60 | 2.60 | NA | NA | NIE | M7 | 15.143 | 17.101 | 16.122 |
| 58 | UPM | 0.5 | 0.60 | 1.40 | NA | NA | NIE | M4 | 6.994 | 8.096 | 7.545 |
| 59 | UPM | 0.5 | 0.60 | 1.60 | NA | NA | NIE | M4 | 7.876 | 8.978 | 8.427 |
| 60 | UPM | 0.5 | 0.60 | 1.90 | NA | NA | NIE | M4 | 9.096 | 10.139 | 9.618 |
| 61 | UPM | 0.5 | 0.60 | 1.95 | NA | NA | NIE | M4 | 10.056 | 11.371 | 10.714 |
| 62 | UPM | 0.5 | 0.60 | 2.10 | NA | NA | NIE | M4 | 10.036 | 11.033 | 10.535 |
| 63 | UPM | 0.5 | 0.60 | 1.10 | NA | NA | NIE | M7 | 9.046 | 10.049 | 9.548 |
| 64 | UPM | 0.5 | 0.60 | 1.30 | NA | NA | NIE | M7 | 11.296 | 13.290 | 12.293 |
| 65 | UPM | 0.5 | 0.60 | 1.60 | NA | NA | NIE | M7 | 13.126 | 15.121 | 14.124 |
| 66 | UPM | 0.5 | 0.60 | 2.10 | NA | NA | NIE | M7 | 15.162 | 16.159 | 15.661 |
| 67 | UPM | 0.5 | 0.60 | 2.30 | NA | NA | NIE | M7 | 17.191 | 17.191 | 17.191 |
| 68 | UPM | 0.5 | 1.32 | 2.70 | NA | NA | NIE | M4 | 16.531 | 18.722 | 17.627 |
| 69 | UPM | 0.5 | 1.32 | 2.80 | NA | NA | NIE | M4 | 18.721 | 21.357 | 20.039 |
| 70 | UPM | 0.5 | 1.32 | 2.90 | NA | NA | NIE | M4 | 17.536 | 18.374 | 17.955 |
| 71 | UPM | 0.5 | 1.32 | 3.00 | NA | NA | NIE | M4 | 16.443 | 18.913 | 17.678 |
| 72 | UPM | 0.5 | 1.32 | 3.10 | NA | NA | NIE | M4 | 14.282 | 16.815 | 15.549 |
| 73 | UPM | 0.5 | 1.32 | 3.30 | NA | NA | NIE | M4 | 16.612 | 19.146 | 17.879 |
| 74 | UPM | 0.5 | 1.32 | 2.60 | NA | NA | NIE | M7 | 30.024 | 30.848 | 30.436 |
| 75 | UPM | 0.5 | 1.32 | 2.70 | NA | NA | NIE | M7 | 27.689 | 30.017 | 28.853 |
| 76 | UPM | 0.5 | 1.32 | 2.80 | NA | NA | NIE | M7 | 33.028 | 33.898 | 33.463 |
| 77 | UPM | 0.5 | 1.32 | 2.90 | NA | NA | NIE | M7 | 32.132 | 34.200 | 33.166 |
| 78 | UPM | 0.5 | 1.32 | 3.00 | NA | NA | NIE | M7 | 34.330 | 36.508 | 35.419 |
| 81 | CEDEX | 1.0 | 1.00 | 2.20 | 1.50 | 0.20 | NIE | M6 | 23.318 | 25.514 | 25.514 |
| 82 | CEDEX | 1.0 | 1.00 | 2.20 | 1.50 | 0.20 | NIE | M6 | 21.520 | 23.557 | 23.557 |
| 83 | CEDEX | 0.5 | 0.40 | 2.20 | 1.50 | 0.10 | NIE | M6 | 7.134 | 8.173 | 7.654 |
| 84 | CEDEX | 0.5 | 0.40 | 1.50 | 1.50 | 0.10 | NIE | M6 | 6.322 | 7.080 | 6.701 |

**Table A1.** *Cont.*

| Test | Laboratory | $H$ (m) | $W$ (m) | $Z_{dss}$ (H:V) | $Z_{uss}$ (H:V) | $l_c$ (m) | IE | Gravel Code | $Q_{f,pre}$ (L·s$^{-1}$) | $Q_{f,pos}$ (L·s$^{-1}$) | $Q_f$ (L·s$^{-1}$) |
|------|------------|---------|---------|-----------------|-----------------|-----------|-----|-------------|---------------------------|---------------------------|----------------------|
| 85 | CEDEX | 1.0 | 1.00 | 2.20 | 1.50 | 0.20 | NIE | M6 | 21.260 | 23.049 | 23.049 |
| 87 | CEDEX | 1.0 | 1.00 | 2.20 | 1.50 | 0.20 | NIE | M6 | 23.288 | 24.773 | 24.031 |
| 88 | CEDEX | 1.0 | 1.00 | 2.20 | 1.50 | 0.20 | NIE | M6 | 21.000 | 21.058 | 21.058 |
| 89 | CEDEX | 1.0 | 1.00 | 1.50 | 1.50 | 0.20 | NIE | M6 | 19.011 | 21.375 | 21.375 |
| 90 | CEDEX | 1.0 | 1.00 | 1.50 | 1.50 | 0.20 | NIE | M6 | 17.146 | 21.144 | 21.144 |
| 91 | CEDEX | 1.0 | 1.00 | 1.50 | 1.50 | 0.20 | NIE | M6 | 17.576 | 21.491 | 21.491 |
| 92 | CEDEX | 1.0 | 1.00 | 1.50 | 1.50 | 0.20 | NIE | M6 | 21.491 | 22.929 | 22.210 |
| 94 | CEDEX | 1.0 | 1.00 | 3.00 | 1.50 | 0.20 | NIE | M6 | 19.933 | 21.144 | 20.539 |
| 95 | CEDEX | 1.0 | 1.00 | 3.00 | 1.50 | 0.20 | NIE | M6 | 19.011 | 19.988 | 19.500 |
| 108 | CEDEX | 0.8 | 1.00 | 1.50 | 1.50 | 0.20 | UF | M7 | 30.265 | 35.233 | 32.749 |
| 109 | CEDEX | 0.229 | 0.40 | 1.50 | 1.50 | 0.06 | UF | M3 | 1.964 | 2.136 | 2.050 |
| 110 | CEDEX | 0.80 | 1.00 | 2.50 | 1.50 | 0.20 | UF | M7 | 39.279 | 45.328 | 42.304 |
| 111 | CEDEX | 0.229 | 0.40 | 2.50 | 1.50 | 0.06 | UF | M3 | 1.495 | 3.798 | 2.647 |
| 112 | CEDEX | 0.80 | 1.00 | 3.50 | 1.50 | 0.20 | UF | M7 | 40.126 | 44.883 | 42.505 |
| 113 | CEDEX | 1.00 | 1.50 | 1.50 | 1.50 | 0.26 | UF | M8 | 60.130 | 103.480 | 81.805 |
| 114 | CEDEX | 1.00 | 1.50 | 2.50 | 1.50 | 0.26 | UF | M8 | 70.150 | 107.800 | 88.975 |
| 123 | CEDEX | 1.00 | 1.00 | 1.50 | 1.50 | 0.20 | CC | M4 | 15.971 | 17.196 | 16.584 |
| 124 | CEDEX | 1.00 | 1.00 | 1.50 | 1.50 | 0.20 | UF | M4 | 17.121 | 19.091 | 18.106 |
| 125 | CEDEX | 1.00 | 1.00 | 2.20 | 1.50 | 0.20 | UF | M4 | 17.121 | 19.011 | 18.066 |
| 126 | CEDEX | 1.00 | 1.00 | 3.00 | 1.50 | 0.20 | UF | M4 | 18.878 | 21.087 | 19.983 |
| 127 | CEDEX | 1.00 | 1.00 | 3.00 | 1.50 | 0.20 | UF | M4 | 18.878 | 21.087 | 19.983 |
| 128 | CEDEX | 1.00 | 1.00 | 1.50 | 1.50 | 0.20 | UF | M4 | 18.878 | 21.144 | 20.011 |
| 129 | CEDEX | 0.229 | 0.40 | 3.50 | 1.50 | 0.06 | UF | M3 | 1.465 | 1.651 | 1.558 |
| 130 | CEDEX | 0.229 | 0.40 | 3.50 | 1.50 | 0.06 | UF | M3 | 2.441 | 2.608 | 2.525 |
| 132 | UPM | 1.0 | 1.32 | 1.90 | NA | NA | NIE | M4 | 27.639 | 30.279 | 28.959 |
| 133 | UPM | 1.0 | 1.32 | 1.60 | NA | NA | NIE | M7 | 44.460 | 46.904 | 45.682 |
| 134 | CEDEX | 1.0 | 1.00 | 3.00 | 1.50 | 0.20 | NIE | M1 | NA | NA | 9.138 |
| 135 | CEDEX | 1.0 | 1.00 | 3.00 | 1.50 | 0.20 | NIE | M1 | NA | NA | 10.336 |
| 136 | CEDEX | 1.0 | 1.00 | 3.00 | 1.50 | 0.20 | NIE | M1 | NA | NA | 10.139 |
| 137 | CEDEX | 1.0 | 1.00 | 3.00 | 1.50 | 0.20 | NIE | M1 | NA | NA | 9.979 |
| 138 | CEDEX | 1.0 | 1.00 | 2.20 | 1.50 | 0.20 | NIE | M1 | NA | NA | 8.957 |
| 139 | CEDEX | 1.0 | 1.00 | 2.20 | 1.50 | 0.20 | NIE | M1 | NA | NA | 7.480 |
| 140 | CEDEX | 1.0 | 1.00 | 1.50 | 1.50 | 0.20 | NIE | M1 | NA | NA | 9.055 |
| 141 | CEDEX | 1.0 | 1.00 | 1.50 | 1.50 | 0.20 | NIE | M1 | NA | NA | 8.616 |
| 142 | CEDEX | 1.0 | 1.00 | 1.50 | 1.50 | 0.20 | NIE | M1 | NA | NA | 9.105 |
| 143 | CEDEX | 1.0 | 1.00 | 1.50 | 1.50 | 0.20 | UF | M1 | NA | NA | 7.945 |
| 144 | CEDEX | 1.0 | 1.00 | 2.20 | 1.50 | 0.20 | UF | M1 | NA | NA | 8.409 |
| 145 | CEDEX | 1.0 | 1.00 | 3.00 | 1.50 | 0.20 | UF | M1 | NA | NA | 8.393 |
| 146 | CEDEX | 1.0 | 1.00 | 1.50 | 1.50 | 0.20 | UF | M1 | NA | NA | 8.159 |
| 147 | CEDEX | 1.0 | 1.00 | 3.00 | 1.50 | 0.20 | CC | M1 | NA | NA | 8.206 |
| 148 | CEDEX | 1.0 | 1.00 | 3.00 | 1.50 | 0.20 | CC | M1 | NA | NA | 8.440 |
| 149 | CEDEX | 1.0 | 1.00 | 2.20 | 1.50 | 0.20 | CC | M1 | NA | NA | 7.156 |
| 150 | CEDEX | 1.0 | 1.00 | 2.20 | 1.50 | 0.20 | CC | M1 | NA | NA | 8.123 |
| 151 | CEDEX | 1.0 | 1.00 | 1.50 | 1.50 | 0.20 | CC | M1 | NA | NA | 7.510 |

**Table A2.** Summary of the average hydraulic pressures $\psi$ at the base of the physical models nº 85, 87, 90, 94, and 95 for a discharge step in the early stages of each test.

| Test | $Z_{dss}$ (H:V) | $Q$ (L·s$^{-1}$) | $x^*$ to Toe | $x^*$ to Crest | $\psi_{average}$ (m) |
|------|-----------------|--------------------|--------------|----------------|------------------------|
| 85 | 2.2 | 5.1 | 0.000 | 1.000 | 0.035 |
| 85 | 2.2 | 5.1 | 0.182 | 0.818 | 0.120 |
| 85 | 2.2 | 5.1 | 0.227 | 0.773 | 0.144 |
| 85 | 2.2 | 5.1 | 0.364 | 0.636 | 0.187 |
| 85 | 2.2 | 5.1 | 0.545 | 0.455 | 0.224 |
| 85 | 2.2 | 5.1 | 0.727 | 0.273 | 0.252 |
| 85 | 2.2 | 5.1 | 0.909 | 0.091 | 0.275 |

**Table A2.** *Cont.*

| Test | $Z_{dss}$ (H:V) | Q (L·s$^{-1}$) | $x^*$ to Toe | $x^*$ to Crest | $\psi_{average}$ (m) |
|---|---|---|---|---|---|
| 87 | 2.2 | 6.7 | 0.000 | 1.000 | 0.036 |
| 87 | 2.2 | 6.7 | 0.182 | 0.818 | 0.129 |
| 87 | 2.2 | 6.7 | 0.227 | 0.773 | 0.155 |
| 87 | 2.2 | 6.7 | 0.364 | 0.636 | 0.205 |
| 87 | 2.2 | 6.7 | 0.545 | 0.455 | 0.244 |
| 87 | 2.2 | 6.7 | 0.727 | 0.273 | 0.273 |
| 87 | 2.2 | 6.7 | 0.909 | 0.091 | 0.296 |
| 90 | 1.5 | 7.1 | 0.000 | 1.000 | 0.009 |
| 90 | 1.5 | 7.1 | 0.267 | 0.733 | 0.120 |
| 90 | 1.5 | 7.1 | 0.333 | 0.667 | 0.149 |
| 90 | 1.5 | 7.1 | 0.533 | 0.467 | 0.203 |
| 90 | 1.5 | 7.1 | 0.800 | 0.200 | 0.257 |
| 94 | 3 | 7.1 | 0.133 | 0.867 | 0.122 |
| 94 | 3 | 7.1 | 0.267 | 0.733 | 0.207 |
| 94 | 3 | 7.1 | 0.300 | 0.700 | 0.224 |
| 94 | 3 | 7.1 | 0.400 | 0.600 | 0.270 |
| 94 | 3 | 7.1 | 0.533 | 0.467 | 0.325 |
| 94 | 3 | 7.1 | 0.667 | 0.333 | 0.343 |
| 94 | 3 | 7.1 | 0.800 | 0.200 | 0.376 |
| 94 | 3 | 7.1 | 0.900 | 0.100 | 0.399 |
| 94 | 3 | 7.1 | 0.933 | 0.067 | 0.408 |
| 95 | 3 | 7.2 | 0.133 | 0.867 | 0.131 |
| 95 | 3 | 7.2 | 0.267 | 0.733 | 0.221 |
| 95 | 3 | 7.2 | 0.300 | 0.700 | 0.239 |
| 95 | 3 | 7.2 | 0.400 | 0.600 | 0.282 |
| 95 | 3 | 7.2 | 0.533 | 0.467 | 0.322 |
| 95 | 3 | 7.2 | 0.667 | 0.333 | 0.349 |
| 95 | 3 | 7.2 | 0.800 | 0.200 | 0.380 |
| 95 | 3 | 7.2 | 0.900 | 0.100 | 0.401 |
| 95 | 3 | 7.2 | 0.933 | 0.067 | 0.410 |

**Table A3.** Summary of the average hydraulic pressures $\psi$ at the base of the physical models nº 87, 94, and 95 for a transversal section roughly located at $x^* = 0.3$ (30% of the base length from the crest).

| Test | $Z_{dss}$ (H:V) | Q (L·s$^{-1}$) | $x^*$ to Toe | $x^*$ to Crest | $\psi_{average}$ (m) |
|---|---|---|---|---|---|
| 87 | 2.2 | 6.7 | 0.727 | 0.273 | 0.274 |
| 87 | 2.2 | 8.9 | 0.727 | 0.273 | 0.370 |
| 87 | 2.2 | 10.7 | 0.727 | 0.273 | 0.407 |
| 87 | 2.2 | 13.5 | 0.727 | 0.273 | 0.437 |
| 87 | 2.2 | 15 | 0.727 | 0.273 | 0.456 |
| 87 | 2.2 | 17.8 | 0.727 | 0.273 | 0.457 |
| 87 | 2.2 | 19.1 | 0.727 | 0.273 | 0.458 |
| 94 | 3 | 12.8 | 0.667 | 0.333 | 0.504 |
| 94 | 3 | 15 | 0.667 | 0.333 | 0.480 |
| 94 | 3 | 19 | 0.667 | 0.333 | 0.415 |
| 94 | 3 | 20 | 0.667 | 0.333 | 0.416 |
| 94 | 3 | 21.2 | 0.667 | 0.333 | 0.428 |
| 95 | 3 | 5 | 0.667 | 0.333 | 0.178 |
| 95 | 3 | 7.1 | 0.667 | 0.333 | 0.240 |
| 95 | 3 | 11.7 | 0.667 | 0.333 | 0.324 |
| 95 | 3 | 13 | 0.667 | 0.333 | 0.342 |
| 95 | 3 | 17.1 | 0.667 | 0.333 | 0.403 |
| 95 | 3 | 21.1 | 0.667 | 0.333 | 0.437 |

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
