# Peer review of "Failure of the Downstream Shoulder of Rockfill Dams Due to Overtopping or Throughflow"

_water, doi:10.3390/w14101624_

Round 1

Reviewer 1 Report

   This manuscript through the laboratory experimental research of 114 physical models studies the failure of the downstream shoulder of highly permeable rockfill subjected to overflow. This result shows the influence of two failure mechanisms and the variable's influence on the downstream slope in this parametric.

  1. In the selection of references, except for ensuring authority, it is necessary to make its source as rich as possible. It is recommended to add more articles from recent 4-5 years as references.;
  2. In order to make readers more convenient to read. It is recommended to insert notes in the manuscript text, not just at the end of the manuscript;
  3. There is a word “ overtopping ” in the manuscript title, but there is no relevant experiment on overtopping in the manuscript text. The test water level reaches the top height of the dam but does not overflow from the top of the dam, and the test phenomenon is closer to the ' piping ';
  4. How are the mechanisms of ' slumping ' and ' particle dragging ' derived from Figure 8?
  5. Table3 \ Table 4 \ Table 5 in the manuscript simply puts the table in the text, but it is not further analyzed;
  6. Please modify the format of the paper according to the requirements of the journal;
  7. The manuscript is lack logicality. There is the suggestion that modifying the logical structure. It is better to put the test results and the relevant mechanism obtained through the test in a similar position;
  8. Zdss did not explain clearly and needed to explain again. Better be shown in the diagram;
  9. Whether the rockfill dam with 40 % porosity exists in reality. Whether it can play its due role. What is the significance of studying this kind of rockfill dam?

   After the author resolved the manuscript according to those recommendations, this manuscript can be accepted for publication.

Reviewer 3 Report

The experimental research of the paper has theoretical significance and application value. But for real dam breaks there are very complex conditions for the upstream water level of the reservoir which affect the discharges of break, such as water level of reservoir decreases quickly during dam break when the reservoir capacity is small enough and the water level of reservoir has no variation during dam break when the reservoir capacity is large enough. So in the paper there should introduce the upstream water level clearly during dam break. So does the downstream water level.

Reviewer 4 Report

In this paper, an extensive laboratory set of tests were carried out to study the failure of the downstream shoulder of highly permeable rockfill subjected to overflow. The experimental research comprised 114 physical models performed varying the following elements: the average size of the uniform gravels, the configuration of the dam, the dam height, the crest length, the downstream slope, the type of impervious element

This topic is interesting. This paper has rigorous structure, clear organization and important innovation. I have some suggestions before it can be published as follows:

  1. Tables 3-5 are attached as appendix.
  2. It is better to compare present experimental data with similar data from references.
